# MAP THE FLOW: REVEALING HIDDEN PATHWAYS OF INFORMATION IN VIDEOLLMS

**Minji Kim**[1*]    **Taekyung Kim**[3*]    **Bohyung Han**[1,2]

[1]ECE & [2]IPAI, Seoul National University
[3]NAVER AI Lab

## ABSTRACT

Video Large Language Models (VideoLLMs) extend the capabilities of vision-language models to spatiotemporal inputs, enabling tasks such as video question answering (VideoQA). Despite recent advances in VideoLLMs, their internal mechanisms on where and how they extract and propagate video and textual information remain less explored. In this study, we investigate the internal information flow of VideoLLMs using mechanistic interpretability techniques. Our analysis reveals consistent patterns across diverse VideoQA tasks: (1) temporal reasoning in VideoLLMs initiates with active cross-frame interactions in early-to-middle layers, (2) followed by progressive video-language integration in middle layers. This is facilitated by alignment between video representations and linguistic embeddings containing temporal concepts. (3) Upon completion of this integration, the model is ready to generate correct answers in middle-to-late layers. (4) Based on our analysis, we show that VideoLLMs retain their VideoQA performance by selecting these effective information pathways while suppressing a substantial amount of attention edges, e.g., 58% in LLaVA-NeXT-7B-Video-FT. These findings provide a blueprint for how VideoLLMs perform temporal reasoning and offer practical insights for improving model interpretability and downstream generalization. Our project page with the source code is available at `https://map-the-flow.github.io`.

## 1 INTRODUCTION

Multimodal large language models (MLLMs) (Chen et al., 2024b;c;a; Liu et al., 2023; 2024a; Wang et al., 2024a) have achieved remarkable success in vision-language tasks by combining powerful auto-regressive language models with vision encoders. Building upon the success of MLLMs, recent efforts have extended these architectures to videos, giving rise to video large language models (VideoLLMs) (Maaz et al., 2024b; Lin et al., 2024; Xu et al., 2024; Wang et al., 2024c) that process spatiotemporal information alongside text. These models have shown promising results on video question answering (VideoQA) tasks, which demand temporal reasoning over multiple frames.

Most prior studies on VideoLLMs have focused on *external* designs of the models, such as scaling video instruction tuning datasets (Li et al., 2023b; Maaz et al., 2024b;a; Li et al., 2024a), key frame selection (Tan et al., 2024; Korbar et al., 2024; Wang et al., 2024b), and compression of input video tokens (Li et al., 2024b; Du et al., 2025; Xu et al., 2024; Zhang et al., 2025b; Jin et al., 2024; Weng et al., 2024; Shen et al., 2025). However, little is known about the *internal* mechanisms of where and how these models extract relevant temporal information from given videos and propagate it through text tokens to generate final answers. Although recent studies on image-based MLLMs (Neo et al., 2025; Zhang et al., 2025c) have identified their structured behaviors for image-text inputs, it remains unclear whether these findings hold in VideoLLMs and what novel capabilities are acquired through video-text alignment beyond image-text pretraining.

In this study, we aim to provide a complete blueprint that reveals the systematic behaviors of VideoLLMs on temporal reasoning tasks, with a focus on the information flow across different layers and modalities. To understand how VideoLLMs generate an answer from a given (video, question)

---

*These authors contributed equally to this work.

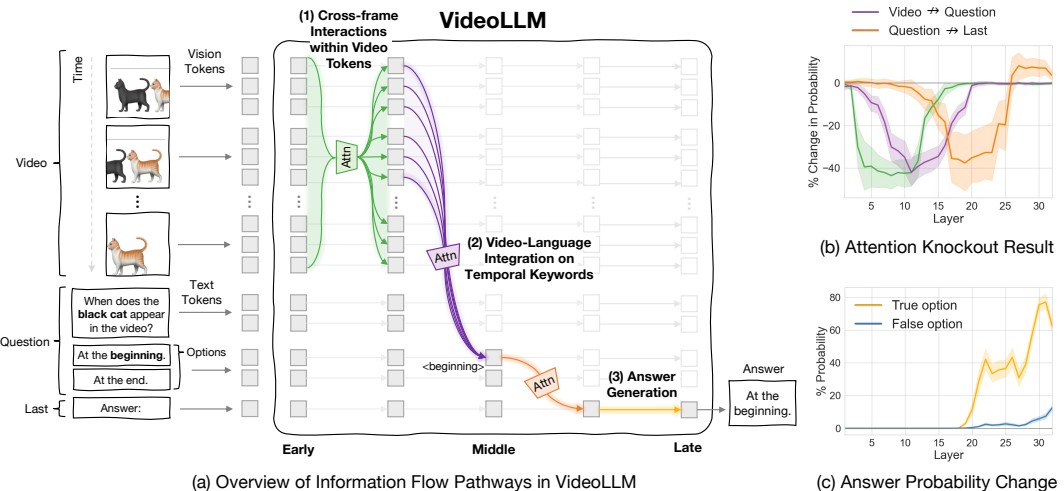

Figure 1: **Summary of our findings on VideoLLMs' information flow. (a)** Temporal reasoning begins with cross-frame interactions within video tokens at early-middle layers [green], followed by video-language integration into temporal keywords in the question [purple]. This information is conveyed to the last token at middle-late layers [orange], where answer generation occurs [yellow]. **(b)** These effective pathways are identified via Attention Knockout, which disconnects attention pairs and tracks the drop in probability of the final answer to quantify their impact. **(c)** Layer-wise answer probability rises immediately after video-language integration, indicating that the model is ready to predict correct answers after the middle layers.

pair, we decompose the temporal reasoning process into several stages and investigate the following key questions: (1) How do VideoLLMs encode spatiotemporal information from the given flattened sequence of video tokens? (2) How are the temporal concepts in the question extracted from video tokens and propagated to text tokens? (3) At what stage does the model become ready to generate an answer? (4) Can we identify effective information flow pathways sufficient to solve VideoQA?

To answer these questions, we take a mechanistic interpretability perspective (Rai et al., 2024; Nanda et al., 2023; Geva et al., 2023) and reverse-engineer the internal computations of VideoLLMs. Our analysis reveals consistent patterns in how VideoLLMs process video-language information across various VideoQA tasks. Our key findings are summarized as follows:

- **Active temporal interaction within video tokens in early-to-middle layers (§3.2):** Temporal reasoning begins by building spatiotemporal representations from video tokens through focused cross-frame attention in early-to-middle layers. Our analysis using Attention Knockout (Geva et al., 2023), which selectively disconnects attention edges to quantify their impact, shows VideoQA instruction tuning from base ImageLLMs distinctly induces this capability.

- **Video-language integration on temporal keywords in middle layers (§3.3):** Analyzing semantic concepts in video tokens through Logit Lens (nostalgebraist, 2020) shows that temporal concepts are emergent among video tokens. Alignment between these representations and temporal keyword embeddings facilitates selective video-language integration over relevant question tokens in early-to-middle layers, which is followed by information converging to the last token in middle-to-late layers.

- **Answer generation at middle-to-late layers (§3.4):** Tracing layer-wise answer probability at the last token reveals that the model is prepared to generate a correct answer immediately after the video-language integration concludes from the middle layers.

- **Effective information flow pathways for solving VideoQA tasks (§3.5):** To validate the above findings, we disable all information pathways except those identified as critical. Evaluation on VideoQA benchmarks shows that the models retain performance comparable to baselines, demonstrating that these effective pathways suffice for accurate answer generation.

Our findings provide a first step in understanding the internal mechanisms of VideoLLMs for temporal reasoning. Code and data are available at https://map-the-flow.github.io.

## 2 PRELIMINARY

### 2.1 VIDEO LARGE LANGUAGE MODELS (VIDEOLLMS)

**Video and instruction tokenization.** Given an input video $V \in \mathbb{R}^{T \times H \times W \times 3}$, where $T$ and $H \times W$ denote the number of frames and the spatial resolution, we patchify each frame into non-overlapping patches of size $p \times p$, resulting in a total of $N_v = T \times \frac{H}{p} \times \frac{W}{p}$ patches. These patches are processed by a vision encoder $f(\cdot)$ to produce a sequence of video tokens $\{\mathbf{v}_i\}_{i=1}^{N_v}$, where $\mathbf{v}_i \in \mathbb{R}^d$. On the other hand, the instruction text $\mathbf{t}$ of length $N_T$ is processed using a tokenizer of the language model component in the VideoLLM, which acts as a lookup table of word embeddings, resulting in a sequence of text tokens $\{\mathbf{t}_i\}_{i=1}^{N_T}$. The video and text tokens are then combined as $[\mathbf{v}_1, ..., \mathbf{v}_{N_v}, \mathbf{t}_1, ..., \mathbf{t}_{N_T}] \in \mathbb{R}^{(N_v + N_T) \times d}$ and fed into the VideoLLM for multimodal processing.

**Causal attention.** For each of $h$ attention heads in transformer layer, $d_h$-dimensional query, key, value representations are computed from the input $\mathbf{x}^{l-1}$ from the previous layer as $\mathbf{q}^l = \mathbf{x}^{l-1}\mathbf{W}_q^l$, $\mathbf{k}^l = \mathbf{x}^{l-1}\mathbf{W}_k^l$, and $\mathbf{v}^l = \mathbf{x}^{l-1}\mathbf{W}_v^l$, where $\mathbf{W}_q^l, \mathbf{W}_k^l, \mathbf{W}_v^l \in \mathbb{R}^{d \times d_h}$ are linear projection matrices. Since VideoLLMs adopt causal attention to preserve the autoregressive nature of generation, the attention output for each head is computed using scaled dot-product attention, which is given by

$$\text{Attention}(\mathbf{q}^l, \mathbf{k}^l, \mathbf{v}^l) = \text{softmax}\left(\frac{\mathbf{q}^l(\mathbf{k}^l)^\top}{\sqrt{d_h}} + \mathbf{M}^l\right)\mathbf{v}^l, \tag{1}$$

where $\mathbf{M}^l$ is a causal mask. The outputs from all heads are concatenated and projected through $\mathbf{W}_o^l \in \mathbb{R}^{d \times d}$ to form the final output of the multi-head attention module at layer $l$.

### 2.2 ATTENTION KNOCKOUT

Attention Knockout (Geva et al., 2023) selectively disables specific attention connections between tokens during inference. This technique allows us to causally trace the contributions of different modalities or frames. By ablating particular attention paths and measuring the impact on predictions, we can uncover the mechanisms by which information propagates through the model, revealing knowledge localization and the functional roles of individual components.

In practice, to prevent information flow from source tokens (e.g., video inputs or earlier frames) to target tokens (e.g., later frames, question, or answer tokens), we set the value of the attention mask $\mathbf{M}^l$ at position $(s, t)$ to $-\infty$ in Eqn. (1), where $s$ and $t$ denote the positions of the source and target tokens, respectively. This replacement ensures that the token at position $t$ cannot attend to the token at position $s$, effectively blocking specific token interactions.

In VideoQA, a model generates an answer $a$ from a given video-question pair $(v, q)$, where the question may contain $n$ options $o = [o_1; o_2; ...; o_n]$. The model initially predicts the answer $a$ with the highest probability $p_{\text{base}}$ at the last token position of the input sequence. We trace the relative probability change in percentage, defined as $((p_{\text{knockout}} - p_{\text{base}})/p_{\text{base}}) \times 100$, where $p_{\text{knockout}}$ is the updated probability for the same answer $a$ derived after intervention.

## 3 INFORMATION FLOW DYNAMICS IN VIDEOLLMS

We investigate the behaviors of VideoLLMs in VideoQA tasks. Our analyses reveal effective information flow pathways of VideoLLMs, leading to three key findings: (1) temporal interactions are primarily established among video tokens in the early-to-middle layers (§3.2); (2) video information is selectively propagated to corresponding temporal reasoning vocabulary tokens and integrated with textual information (§3.3); and (3) answer generation emerges near the completion of the video-language integration and progresses through the mid-to-late layers (§3.4). We further discuss the impact of pruning ineffective information flow pathways of VideoLLMs on the VideoQA performance (§3.5). We present the analysis setup (§3.1) and provide a detailed discussion in the following sections. Our analyses focus on multiple-choice VideoQA samples for structured evaluation, with open-ended question extensions in the Appendix (§B).

Table 1: **Overview of tasks and data for our analyses.** We adopt five tasks from TVBench (Cores et al., 2024), a multiple-choice VideoQA benchmark covering diverse temporal reasoning types.

| Task | Reasoning Type | Question Example | Option Example |
|------|---------------|------------------|----------------|
| Action Antonym | Action recognition Sequential ordering | What is the action being performed in the video? | 2 temporally opposite actions e.g., Wear jacket.; Take off jacket. |
| Action Sequence | Action recognition Temporal localization | What did the person do first? | 2 actions that actually happened on the video e.g., Put down the blanket.; Took the towel. |
| Scene Transition | Scene recognition Sequential ordering | What's the right option for how the scenes in the video change? | From {scene1} to {scene2}. From {scene2} to {scene1}. |
| Moving Direction | Moving object properties | Which direction does the gray cube move in the video? | Down and to the right.; Down and to the left. Up and to the right.; Up and to the left. |
| Object Count | Moving object properties Temporal localization | How many metal objects are moving when the video begins? | 0; 1; 2; 3 |

## 3.1 EXPERIMENTAL SETUP

**Tasks and data.** We construct our data by selecting five tasks from TVBench (Cores et al., 2024), a multiple-choice VideoQA benchmark designed to evaluate temporal understanding while minimizing static-scene bias. As shown in Table 1, our data contains tasks requiring reasoning over diverse attributes under temporally challenging situations. To guarantee the validity of our causal tracing, we restrict the analysis to instances where the model provides the correct answer. This filtering step ensures our study focuses on samples where the model successfully reasons about visual and temporal content, thereby eliminating noise from random guesses or fundamental misunderstandings. Extended analyses covering open-ended QA, long-form video understanding, and spatial reasoning are provided in the Appendix.

**Models.** We study the behavior of MLLMs that are fine-tuned with video instruction tuning. Specifically, we focus on models originally trained on static image-text data and later fine-tuned on video datasets to analyze unique properties learned during the video instruction tuning procedure. To this end, we fine-tune LLaVA-NeXT-7B (Liu et al., 2024b) with VideoChat2-IT (Li et al., 2024a) for 3 epochs and use this model for the analysis in our main paper. For convenience, we refer to this model as LLaVA-NeXT-7B-Video-FT. For both training and inference, we use 8-frame sampling with 144 tokens per frame. We further extend our analyses on other VideoLLMs with diverse architectures, including LLaVA-NeXT-13B-Video-FT, Mini-InternVL-4B-Video-FT (Gao et al., 2024), and VideoLLaMA3-7B (Zhang et al., 2025a) in the Appendix.

## 3.2 TEMPORAL INTERACTION WITHIN VIDEO TOKENS

To successfully perform VideoQA, VideoLLMs must aggregate information distributed across temporal dimensions and produce the correct answer at the final token position. This subsection investigates the internal mechanisms by which VideoLLMs encode spatiotemporal features from a flattened sequence of video tokens.

**Training with VideoQA data boosts cross-frame interactions in the early-to-middle layers.** While ImageQA tasks, such as object identification, can often be solved by localizing specific regions at the token level, VideoQA presents a distinct challenge. In VideoQA, visual information is distributed across a sequence of frames, requiring the model to integrate spatiotemporal cues to capture essential temporal concepts. To investigate how these disparate task requirements shape the internal mechanisms of MLLMs, we compare Attention Knockout results between a model trained exclusively on image data (LLaVA-NeXT-7B) and its video-fine-tuned counterpart (LLaVA-NeXT-7B-Video-FT). Specifically, for each layer $l$, we inhibit vision tokens from attending to tokens in preceding frames within a $k(=9)$ layer window[1] centered at $l$, and measure the percentage relative change in answer prediction probability. As shown in Figure 2, disrupting cross-frame interactions in

---

[1] We observed that blocking too narrow a range allows information to bypass the intervention via residual connections, whereas wider windows robustly induce a more significant degradation in prediction probability. Consequently, we adopt a window size of $k = 9$ following (Geva et al., 2023); further sensitivity analyses are provided in §G.6.

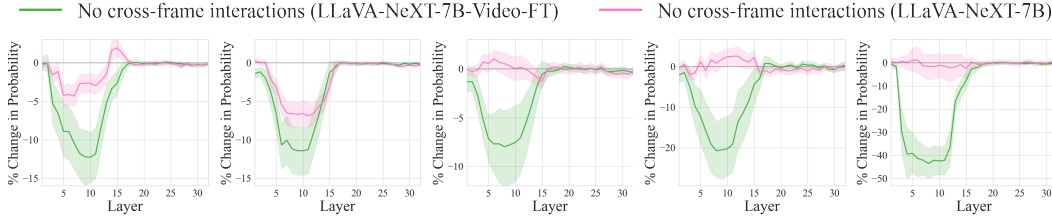

(a) Action Antonym  (b) Action Sequence  (c) Scene Transition  (d) Moving Direction  (e) Object Count

Figure 2: **Change in prediction probability when disconnecting cross-frame attention edges.** Blocking cross-frame interactions in early-to-middle layers significantly harms LLaVA-NeXT-7B-Video-FT's prediction, while LLaVA-NeXT-7B remains mostly unaffected.

Table 2: **Impact of cross-frame attention on answer generation.** We block cross-frame attention in the first half of the total layers and measure the resulting accuracy drop. Answers in the third column are taken from open-ended responses from each case. Without cross-frame attention, the model generates incorrect or even opposite answers to the given videos.

| Task | Acc Drop | Answer Example |
|------|----------|----------------|
| Action Antonym | −24.1% | *Baseline*: The action being performed in the video is to  stand up . |
| | | *Knockout*: The action being performed in the video is to  sit on a chair . |
| Action Sequence | −20.2% | *Baseline*: The action the person is doing first is to  open the plastic bag . |
| | | *Knockout*: The action the person is doing first is to  put a bag in the microwave . |
| Scene Transition | −18.0% | *Baseline*: The scene in the video changes  from the bedroom to the street . |
| | | *Knockout*: The scene in the video changes  from the street to a different location . |
| Moving Direction | −44.8% | *Baseline*: The purple sphere moves  to the right  in the video. |
| | | *Knockout*: The purple sphere moves  to the left  in the video. |
| Object Count | −60.8% | *Baseline*: The number of moving objects is  zero  when the video begins. |
| | | *Knockout*: The number of moving objects is  three  when the video begins. |

the early-to-middle layers consistently degrades the performance of the VideoLLM across all tasks. In contrast, the ImageLLM does not exhibit such layer-wise sensitivity in most scenarios. These results suggest that VideoQA fine-tuning explicitly fosters robust cross-frame dependency within the model's internal representations.

**Impact of early-stage temporal interactions on answer generation.**    To investigate the extent to which early-to-middle layer temporal interactions contribute to the final answer generation, we inhibit cross-frame attention in the first half of the model (i.e., layers 1–16) and evaluate the resulting impact on baseline performance. As shown in Table 2, this intervention results in accuracy drops of at least 18% among samples that were correctly predicted under full causal attention. The qualitative analysis in the third column—displaying the model's open-ended responses—reveals that the intervention leads to incorrect or even contradictory answers across all tasks. These findings imply that VideoLLMs rely heavily on cross-frame interactions during the early stages of processing to effectively reason about temporal events.

### 3.3 VIDEO-LANGUAGE INTEGRATION ON TEMPORAL REASONING KEYWORDS

Having demonstrated that early-layer cross-frame interactions facilitate the formation of spatiotemporal representations, we now examine how this visual information integrates with textual tokens. To establish a baseline, we trace the overall video-to-language information flow in Fig. 3, which reveals that VideoLLMs operate through a structured pathway originating from video tokens, passing through question tokens, and culminating at the final token. Building on this observation, we investigate the mechanisms by which VideoLLMs selectively propagate spatiotemporal information via specific temporal reasoning keywords.

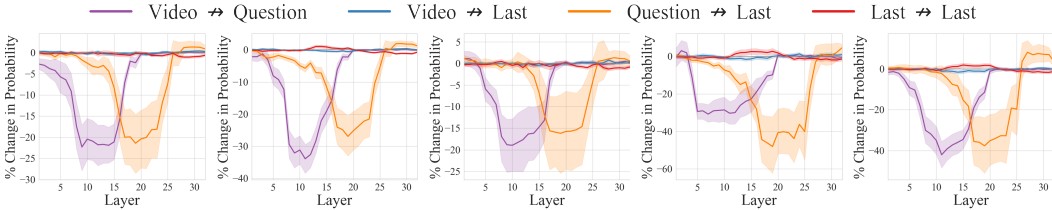

(a) Action Antonym  (b) Action Sequence  (c) Scene Transition  (d) Moving Direction  (e) Object Count

Figure 3: **Overall cross-modal information flow in VideoLLMs.** We analyze changes in the prediction probability when intervening on attention edges between video, question, and last token (i.e., the starting position for answer generation), following the protocol of (Zhang et al., 2025c). Information from the video tokens is conveyed to the question tokens in the early-to-middle layers, followed by the transfer of information from the question tokens to the last token in the middle-to-late layers. *Source ↛ Target* indicates blocking attention edges from source tokens to the target tokens.

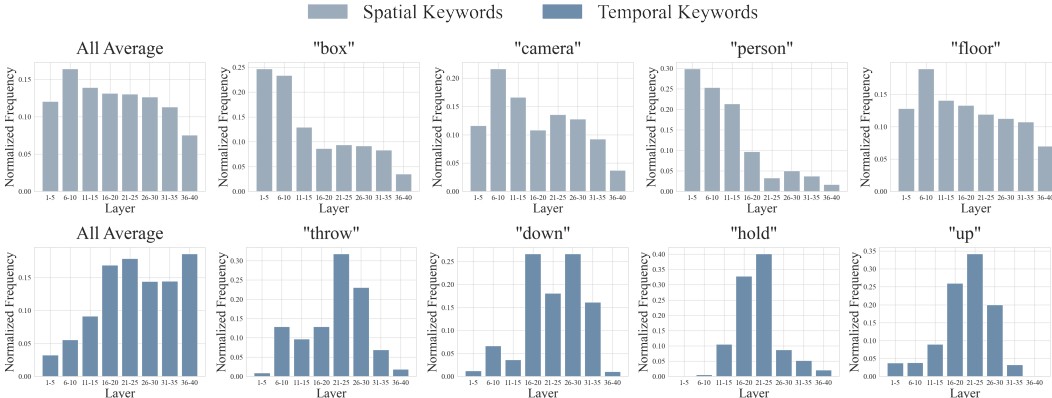

Figure 4: **Normalized frequency of spatial and temporal keywords extracted from video tokens via Logit Lens.** Spatial concepts start to appear in the very early layers, whereas temporal concepts develop later in the middle layers onward. The full list of keywords is shown in Table D.

**Emergence of temporal concepts in video tokens.** Which semantic concepts are extracted from video tokens, and how do they emerge across layers? To answer this, we employ Logit Lens (nostalgebraist, 2020) to trace the evolution of latent concepts across layers. Specifically, we project the hidden states of video tokens at all layers through the language model head to obtain logits, then count the occurrence of spatial and temporal keywords to examine their distribution across layers. We use LLaVA-NeXT-13B-Video-FT on videos from the Action Sequence task, with spatial and temporal keywords parsed from the question prompts. Figure 4 shows that both spatial and temporal concepts are captured in video tokens, but with distinct emergence patterns: spatial concepts start to appear in very early layers, while temporal concepts develop in middle layers onward.

To further validate the disparity in emergence timelines, we visualize the evolution of logit distributions for these key vocabularies within a video in Fig. 5. Specifically, we map the patch token positions associated with spatial and temporal keywords identified using Logit Lens across the early, middle, and late layers. Our findings reveal a sequential allocation of token positions: the model first grounds spatial concepts (e.g., salient entities or attributes) at foreground tokens in early layers, and later utilizes the remaining token positions to represent temporal dynamics, rather than overriding the already stabilized spatial representations.

**Video-language alignment enables selective spatiotemporal propagation.** How are the emergent concepts in videos propagated through text tokens? We analyze the propagation of spatiotemporal information to question tokens and compare it against the propagation of static visual information. We qualitatively show two aspects: (1) temporal visual information is aligned with temporal concept vocabularies, and (2) such alignment emerges specifically through cross-frame interactions.

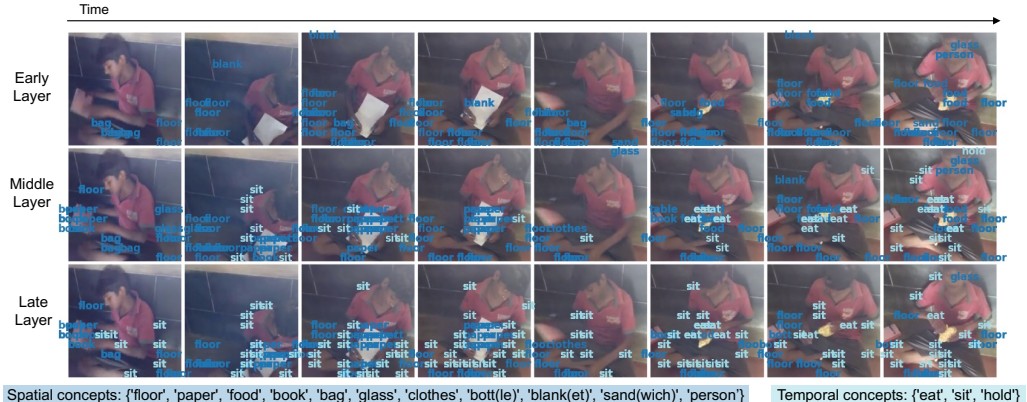

Figure 5: **Visualization of spatial and temporal concepts extracted by Logit Lens.** To investigate the emergence timing and positional distribution of concepts, we visualize the vocabularies extracted from video tokens across early, middle, and late layers. Spatial concepts tend to localize on salient regions early, and temporal concepts then emerge mainly on the remaining tokens rather than displacing already stabilized spatial tokens. The full visualization across all layers is shown in Fig. F.

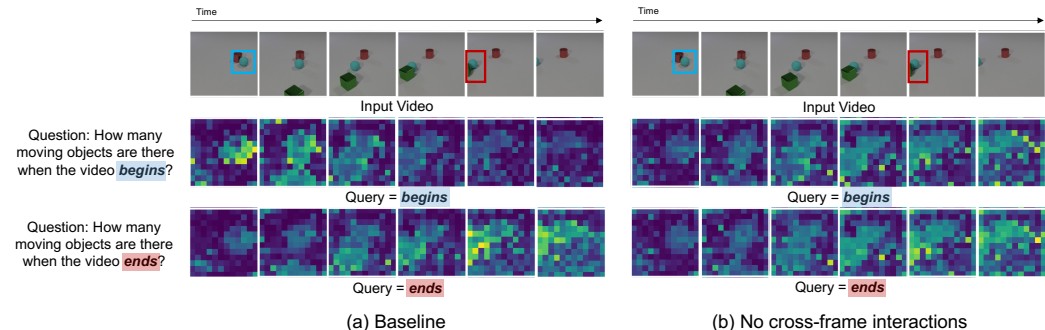

Figure 6: **Visualization of video-to-question attention maps.** Queries are "begins" and "ends" question tokens; keys are video tokens. (a) With spatiotemporal interactions, each question token attends to semantically relevant regions: "begins" focuses on the blue sphere at the start, while "ends" focuses on the blue sphere and green square at the end. (b) When temporal interactions among video tokens are blocked, video-text alignment fails, and text tokens instead attend to positionally proximate regions rather than semantically relevant ones.

To this end, we compare video-to-question attention of different temporal concept words in the question (e.g., "begins", "ends") before and after masking cross-frame attention. As illustrated in Fig. 6(a), when the standard cross-frame attention is maintained, attention maps highlight the semantically relevant temporal segment of the video corresponding to the temporal meanings of the words "begins" and "ends". This demonstrates that spatiotemporal interactions enable the selective exploitation of semantically crucial information throughout the video, allowing question tokens to focus on the most relevant regions across both spatial and temporal dimensions. In contrast, when this cross-frame attention is blocked in Fig. 6(b), the model fails to associate temporal concept vocabulary with relevant video content and instead exhibits positional bias based on positional proximity toward question tokens. These findings indicate that VideoLLMs implicitly learn to align spatiotemporal representations with linguistic embeddings corresponding to temporal concepts.

**Temporal keywords serve as core checkpoints in video-text information flow pathways.** Interleaving the two previous findings raises a natural question of how video information is propagated to the last token via temporal reasoning keywords. However, directly tracing keyword-to-keyword pathways is non-trivial, as the relevant keywords vary across questions. Instead, we analyze the information flow through the options, leveraging the fact that multiple-choice options consistently act as temporal keywords in VideoQA. We segment the full question prompt into fine-grained components: the non-option question (e.g., *"What is the action being performed in the video?"*), the true option

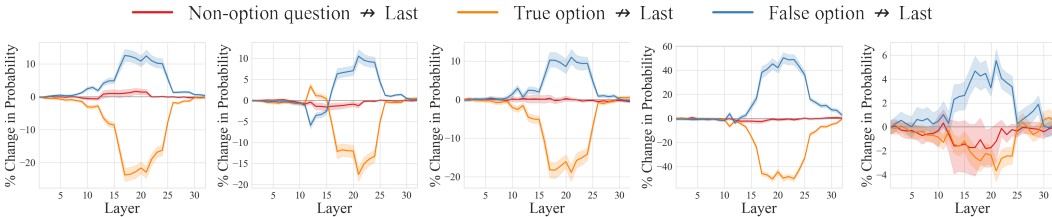

(a) Action Antonym   (b) Action Sequence   (c) Scene Transition   (d) Moving Direction   (e) Object Count

Figure 7: **Change in the prediction probability when intervening on attention edges from different parts of the question tokens to the last token.** *Source ↛ Target* indicates blocking attention edges from source positions to the target positions. Most of the information flowing to the last token in the middle-to-late layers derives from the true option tokens, rather than the broader context in the non-option question. Note that the observed probability rise in *false option ↛ last* is likely because removing the false option makes the task easier to solve.

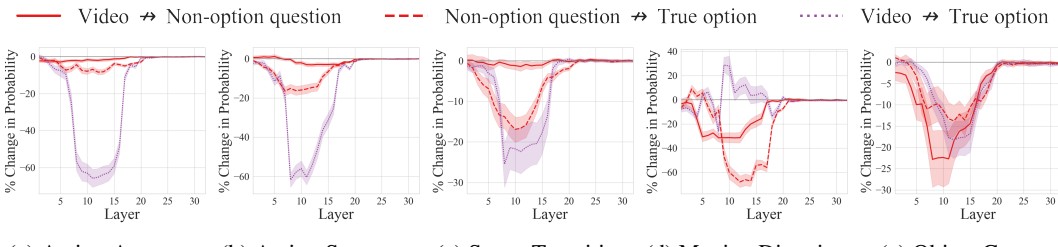

(a) Action Antonym   (b) Action Sequence   (c) Scene Transition   (d) Moving Direction   (e) Object Count

Figure 8: **Change in the prediction probability when intervening on attention edges to the true option position.** Information from video tokens consistently converges to the true option tokens in early-to-middle layers, while routing to non-option question tokens varies depending on the task.

(e.g., *"(A) Wear jacket"*), and the false option (e.g., *"(B) Take off jacket"*). We then examine where the last token primarily derives information from. Figure 7 shows that non-option question tokens contribute little direct flow to the last token, whereas information from the true option is propagated in the middle-to-late layers. This indicates that video-language integration completes at the option tokens containing temporal keywords.

However, the pathways by which video information reaches option tokens may be question-dependent, as temporal keywords are embedded differently across questions. To verify this, we decompose the flow toward the true option into two routes: (1) a direct route (*video → true option*) and (2) an indirect route (*video → non-option question → true option*), and compare their layer-wise patterns across tasks. Figure 8 reveals divergent, task-specific strategies. In Action Antonym, Action Sequence, and Scene Transition, video information is primarily transferred via the direct route (purple), with minor contributions from non-option question tokens. Conversely, in Moving Direction and Object Count, information regarding the queried target object is first routed to the non-option question (red) and then propagated to the true option (red dotted line).

These results indicate that temporal keywords function as information integration checkpoints, with their specific pathways adapting to the underlying prompt structure. Our extended analysis of open-ended answer generation (see Appendix B) further supports this conclusion; we observe that the model dynamically establishes these checkpoints around temporal keywords (e.g., verbs) to facilitate the propagation of video information critical for response generation.

## 3.4 ANSWER GENERATION BEHAVIOR AT MIDDLE-TO-LATE LAYERS

We further examine the role of layers beyond the information propagation stage. Regarding the prior study (Zhang et al., 2025c) that the last layers of MLLMs primarily focus on linguistic completion, we trace the evolution of the answer generation process. Specifically, we probe the layer-wise hidden representations at the last token position to monitor the probabilities for both the true and false

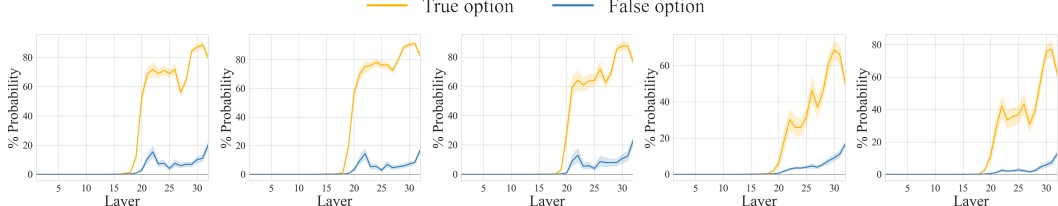

(a) Action Antonym    (b) Action Sequence    (c) Scene Transition    (d) Moving Direction    (e) Object Count

Figure 9: **Layer-wise prediction probability for true and false options in the last token position.** The probability for the true option starts to rise immediately after the middle layers.

Table 3: **Impact of effective flow pathways on performance in TVBench and TOMATO.** The number of attention edges is the count of valid (query, key) pairs over all attention layers. When we enable attention only for effective pathways, VideoQA performance is retained across diverse tasks and models even though suppressing a substantial portion of attention edges.

| Model | # Video Tokens | Attention Type | # Attention Edges | TVBench | TOMATO |
|---|---|---|---|---|---|
| LLaVA-NeXT-7B-Video-FT | $8 \times 12 \times 12$ | Full causal attention | 25.7M (100%) | 51.5 | 30.2 |
| | | Effective pathways | 10.8M (42%) | 51.2 | 29.2 |
| | | Random blocking | 10.8M (42%) | 40.1 | 23.1 |
| LLaVA-NeXT-13B-Video-FT | $8 \times 12 \times 12$ | Full causal attention | 32.2M (100%) | 55.1 | 27.2 |
| | | Effective pathways | 14.3M (37%) | 54.6 | 27.4 |
| | | Random blocking | 14.3M (37%) | 41.5 | 23.8 |
| Mini-InternVL-4B-Video-FT | $8 \times 16 \times 16$ | Full causal attention | 74.6M (100%) | 56.0 | 32.2 |
| | | Effective pathways | 29.6M (40%) | 56.0 | 31.2 |
| | | Random blocking | 29.6M (40%) | 41.0 | 25.9 |
| VideoLLaMA3-7B | $8 \times 12 \times 12$ | Full causal attention | 19.9M (100%) | 55.2 | 28.0 |
| | | Effective pathways | 11.4M (58%) | 57.2 | 28.7 |
| | | Random blocking | 11.4M (58%) | 22.2 | 13.9 |

options. Figure 9 shows that the prediction probability for the true option rises abruptly starting around the 20th layer, which coincides with the completion of the video-to-question information flow. Furthermore, the true option rapidly dominates the output probability rather than exhibiting a gradual phase of competition among all options. This suggests that the final decision for a correct answer is contingent upon the success of the video-to-language propagation in the preceding middle layers.

## 3.5 DOMINANT CONTRIBUTION OF EFFECTIVE INFORMATION FLOW TO VIDEOQA

We have discovered the effective information flow pathways of the temporal reasoning process within VideoLLMs. This observation raises a natural question regarding the contribution of these pathways to overall VideoQA performance. To assess the impact of effective information pathways, we conduct a quantitative analysis by evaluating VideoLLMs on VideoQA benchmarks after retaining only the identified effective token interactions, while disabling all others[2]. Table 3 summarizes the performance on TVBench (Cores et al., 2024) and TOMATO (Shangguan et al., 2024) benchmarks. Although restricting attention to these effective pathways prunes a substantial portion of attention edges (e.g., retaining only 42% of original edges in LLaVA-NeXT-7B-Video-FT), it results in only a marginal degradation in accuracy across both benchmarks. Conversely, randomly sparsifying the same proportion of attention edges leads to a severe performance drop. These findings validate our characterization of the effective information flow pathways and their role in VideoLLM inference.

---

[2]The layer ranges for effective pathways are empirically determined based on the patterns observed in Sections 3.2–3.4, specifically targeting intervals where blocking induced significant probability drops (see Table E for details). We enable *cross-frame interactions* (e.g., L6–15) and *video → question* flows (e.g., L6–20) in early-to-middle layers, followed by *question → last* token transitions in middle-to-late layers (e.g., L16–25). In contrast, *video → last* and *last → last* connections are disabled across all layers. Furthermore, flows directed toward *video* and *question* tokens are blocked in later layers as they no longer contribute to the final output.

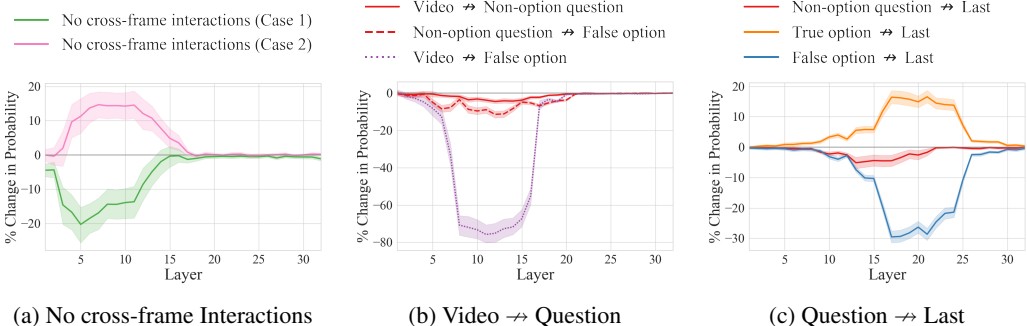

Figure 10: **Information flow in failed VideoQA samples.** Tracing the probability change of the wrong answer in failed samples shows that (a) errors stem from spurious cross-frame attentions (Case 1) or static bias (Case 2) in the early layers. In contrast, (b–c) cross-modal integration from middle layers shows a similar pattern to the successful case, indicating that the failure arises from earlier spatiotemporal representation building rather than a collapse of the cross-modal integration pathways.

## 4 DISCUSSION

**Failure case analysis.** To mechanistically understand why the model makes incorrect predictions, we analyze how the prediction probability of the wrong answer changes in failed VideoQA samples in the Action Antonym task. Figure 10(b–c) illustrates that the cross-modal flow patterns routing false options in failed samples are similar to those routing correct options in successful samples. This suggests that failures may stem from inaccuracies established during the initial video representation stage rather than a breakdown in the subsequent integration pathways. In contrast, disabling cross-frame interaction (Fig. 10(a)) reveals two distinct failure modes. In Case 1 (green), the incorrect option probability decreases upon disabling cross-frame attention, suggesting that the erroneous signal carried by these edges was a primary cause of the model's misprediction. In Case 2 (pink), however, the incorrect option probability increases, indicating a reliance on static bias where the model defaults to uninformative static information cues in the absence of temporal context.

**Future applications of our findings.** First, our analysis reveals that current VideoLLMs rely on a highly concentrated set of dominant information pathways, suggesting the need for pathway regularization during training. Suppressing dominant pathways could encourage the model to explore alternatives (Zehui et al., 2019; Li et al., 2023a), thereby better utilizing its representational capacity. Second, our findings of concept emergence and failure cases highlight the importance of establishing temporal concepts in earlier layers while reducing static-scene bias. This implies that training objectives or architectural constraints promoting the early formation of visual representations (Nikankin et al., 2025; Neo et al., 2025) may improve the robustness of temporal reasoning. Finally, our identification of effective information flow ranges indicates that token interactions beyond these ranges contribute marginally to the final decision. This discovery paves the way for early-exit strategies (Elbayad et al., 2020; Schuster et al., 2022; Bae et al., 2023) that adaptively truncate redundant computations, substantially reducing inference overhead while preserving accuracy.

## 5 CONCLUSION

We have conducted a comprehensive mechanistic analysis to understand the spatial and temporal dynamics of information extraction and propagation within VideoLLMs for VideoQA tasks. Our study reveals that temporal reasoning is initiated by the formation of spatiotemporal representations of video tokens during the early-to-middle layers, driven by active cross-frame interactions. This processed visual information is then selectively transferred to semantically aligned temporal keywords within the question. Furthermore, we observe that the critical information required for final decision-making is conveyed to the final token during the middle-to-late layers, where it culminates in answer generation. These sparse pathways prove sufficient for effectively solving VideoQA tasks. Our findings provide practical insights into the internal working mechanisms of VideoLLMs and pave the way for future research into their interpretability and architectural generalization.

REPRODUCIBILITY STATEMENT

To ensure reproducibility, we provide comprehensive implementation details in Appendix H, including model architectures, training and inference strategies, and experimental configurations. Our Attention Knockout and Logit Lens analyses are implemented based on the public codebase from (Geva et al., 2023). All experiments use publicly available datasets [TVBench (Cores et al., 2024), TOMATO (Shangguan et al., 2024), VCGBench (Maaz et al., 2024b), LongVideoBench (Wu et al., 2024), Video-MME (Fu et al., 2024)] and models [LLaVA-NeXT (Liu et al., 2024b), Mini-InternVL (Gao et al., 2024), VideoLLaMA3 (Zhang et al., 2025a)]. Our complete code and models are available at the project page.

ACKNOWLEDGMENTS

This work was partly supported by the National Research Foundation of Korea (NRF) grant [RS-2022-NR070855] and by the Institute of Information & communications Technology Planning & Evaluation (IITP) grants [No.RS-2022-II220959 (No.2022-0-00959), (Part 2) Few-Shot Learning of Causal Inference in Vision and Language for Decision Making, No.RS-2021-II211343, Artificial Intelligence Graduate School Program (Seoul National University)] funded by the Korean government (MSIT).

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

APPENDIX

## A  RELATED WORK

**Video Large Language Models (VideoLLMs).**    Research on video understanding has increasingly focused on leveraging image-level pre-trained MLLMs by fine-tuning them for video-language tasks such as video question answering, video captioning, and video conversation. To improve the temporal reasoning ability of VideoLLMs, the majority of the existing studies have concentrated on the *external* aspects of VideoLLMs, such as scaling video instruction tuning datasets (Li et al., 2023b; Maaz et al., 2024b;a; Li et al., 2024a), selecting key frames (Tan et al., 2024; Korbar et al., 2024; Wang et al., 2024b), retaining memory banks (Song et al., 2024; He et al., 2024), and compressing the input video tokens (Li et al., 2024b; Du et al., 2025; Xu et al., 2024; Zhang et al., 2025b; Jin et al., 2024; Weng et al., 2024; Shen et al., 2025). In contrast, our study focuses on the *inner* mechanism of VideoLLMs and investigates how temporal reasoning occurs.

**Mechanistic interpretability of multimodal models.**    Mechanistic interpretability (Rai et al., 2024; Nanda et al., 2023; Geva et al., 2023) is an emerging area that seeks to understand neural networks by reverse-engineering their internal computations. Recently, several studies (Palit et al., 2023; Yu & Ananiadou, 2024; Cohen et al., 2025; Basu et al., 2024; Neo et al., 2025; Zhang et al., 2025c) have applied mechanistic interpretability techniques to MLLMs to explain their inner mechanisms. (Basu et al., 2024) focused on how information is stored and retrieved from the model parameters of MLLMs. On the other hand, (Neo et al., 2025) examined the object identification task and explored how the object-level information flows and emerges. Most recently, (Zhang et al., 2025c) systematically studied cross-modal information flow in MLLMs, revealing that many models exhibit a single-stream information transfer pattern from the visual to the linguistic modality. Building upon these methodologies, we extend attention-based knockout and layer-wise logit probing techniques to VideoLLMs to understand temporal reasoning mechanisms.

**Video Question Answering (VideoQA) benchmarks.**    VideoQA benchmarks have evolved to validate temporal understanding for VideoLLMs. VCGBench (Maaz et al., 2024b) and Video-MME (Fu et al., 2024) provide wide coverage of aspects that appear within videos, establishing strong general-purpose baselines. TVBench (Cores et al., 2024) is designed to require true temporal reasoning and to discourage single-frame shortcuts. TempCompass (Liu et al., 2024c) evaluates multiple temporal aspects through varied VideoQA formats to measure time-aware perception. Vinoground (Zhang et al., 2024) uses counterfactual short video pairs to assess fine-grained temporal distinctions beyond static cues. TemporalBench (Cai et al., 2024) focuses on detailed temporal dynamics such as event order frequency and change patterns. TOMATO (Shangguan et al., 2024) introduces expert-written tasks that force reasoning about event evolution across frames. MotionBench (Hong et al., 2025) targets fine-grained motion comprehension using motion-centric questions. LongVideoBench (Wu et al., 2024) evaluates long-form interleaved video language understanding, where models must retrieve relevant moments and maintain evidence over extended contexts. In this study, we primarily focus our analysis on TVBench, TOMATO, VCGBench, LongVideoBench, and Video-MME.

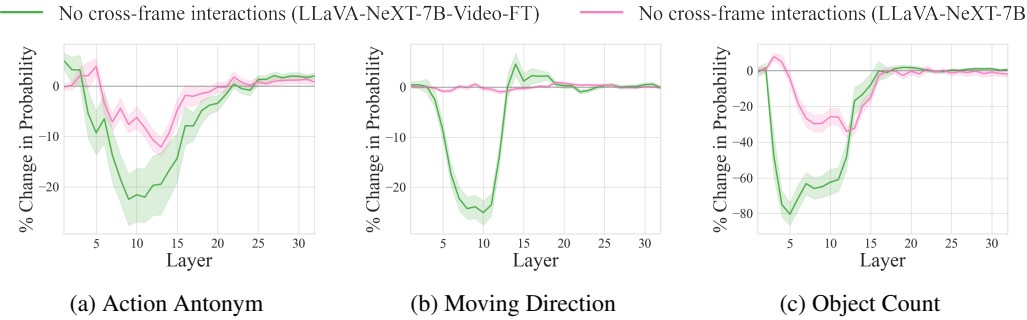

(a) Action Antonym  (b) Moving Direction  (c) Object Count

Figure A: **Change in prediction probability in open-ended QA format when disconnecting cross-frame attention edges.** LLaVA-NeXT-7B-Video-FT shows a stronger correlation with cross-frame interactions and the final answer probability compared to LLaVA-NeXT-7B.

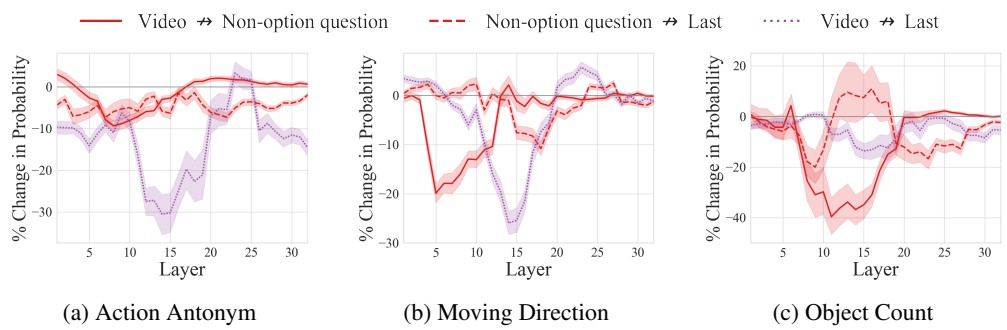

(a) Action Antonym  (b) Moving Direction  (c) Object Count

Figure B: **Change in the prediction probability in open-ended QA format when intervening on attention edges between video, non-option question, and last token.** In the absence of explicit temporal keywords in the open-ended format, the last position itself serves as a checkpoint for video-text integration in the middle layers.

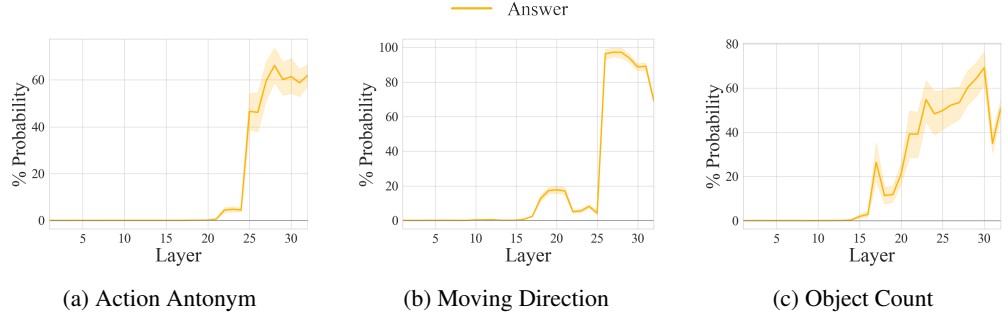

(a) Action Antonym  (b) Moving Direction  (c) Object Count

Figure C: **Layer-wise prediction probability for ground truth answers in open-ended QA format in the last token position.** The probability for the ground truth starts to rise immediately after the middle layers.

## B  ANALYSIS ON OPEN-ENDED VIDEOQA

In the main text, we demonstrated that temporal keywords in the prompt serve as critical checkpoints in information flow pathways for multiple-choice QA tasks. In open-ended VideoQA, however, the prompt lacks explicit keywords that correspond to the ground-truth answer. Consequently, the information flow to the last token may differ, as the model must independently generate answer vocabularies rather than simply selecting from the given options. In this section, we investigate whether this difference in prompt format alters the established mechanisms of information propagation. We first analyze single-token generation (§B.1) and extend our investigation to multiple-token generation scenarios (§B.2).

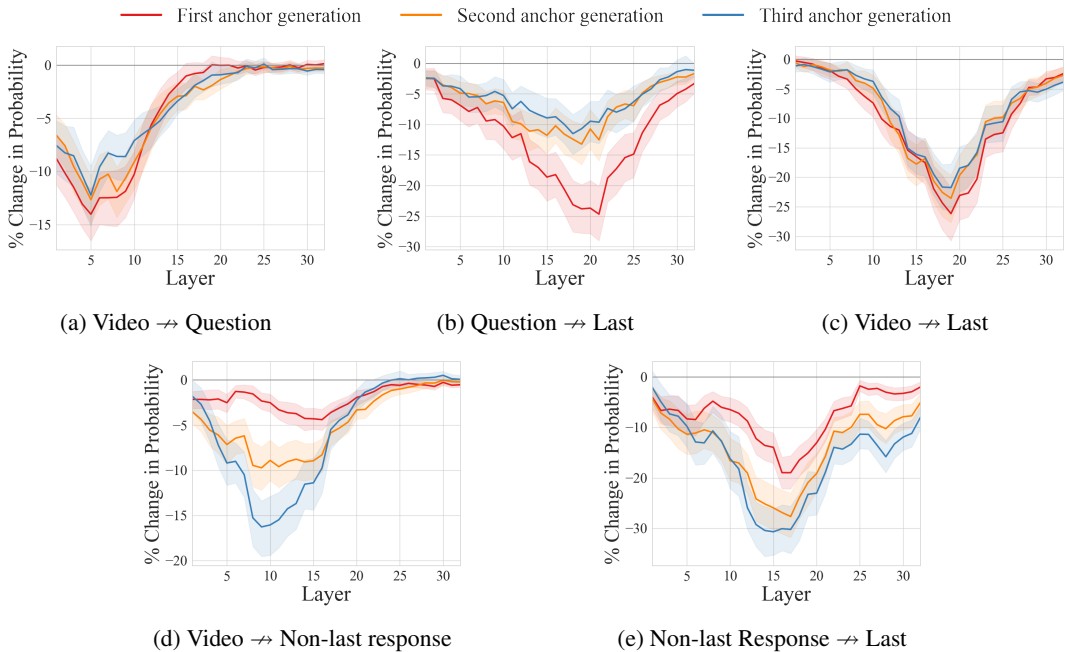

Figure D: **Open-ended generation analysis on VCGBench.** We set verbs in the generated response as potential semantic anchors containing temporal vocabulary and measure a probability drop at generating each anchor when intervening attention edges from *Source ↛ Target* tokens.

### B.1 SINGLE TOKEN GENERATION

**Open-ended analysis setup.** The input prompt formats in TVBench are modified by removing the options and adopting a sentence completion style. For example: *"USER: <video> USER: Question: Which direction does the gray cube move in the video? ASSISTANT: The gray cube moves to the ___."* To avoid ambiguity, we select tasks where the first tokenized sub-word of the model's possible answer is relatively constrained, such as Action Antonym, Moving Direction, and Object Count. We adopt LLaVA-NeXT-7B-Video-FT and LLaVA-NeXT-7B for this analysis.

**Active temporal interaction in open-ended VideoQA.** We examine the impact of temporal interaction within video tokens in open-ended VideoQA. Using the same attention-blocking setup as in previous evaluations, we observed that disabling cross-frame interactions in early-to-middle layers leads to a significant decrease in answer probability even in the open-ended questions answering problems, as shown in Fig. A. These results suggest that active temporal interaction is a general mechanism leveraged by VideoLLMs, regardless of the format of the question answering problems.

**Video-language integration.** Unlike multiple-choice question answering, open-ended question answering does not explicitly provide candidate answers. Consequently, the input text lacks explicit temporal reasoning keywords that directly reference temporal information in the video. We hypothesize that, in the absence of true option tokens, the last token itself becomes the core checkpoint for video-language integration. To this end, we examine the information flow from the video and question tokens to the last token. Figure B shows two different routes: *video → last* (purple lines) and *video → non-option question → last* (red lines). While video information may first pass through question tokens, the final integration converges at the last token. This behavior aligns with what we have observed in multiple-choice tasks, where video and language information merge at a core checkpoint, although this checkpoint shifts to the last token position in the open-ended case.

**Answer generation.** We observe that answer generation occurs at middle-to-late layers also for open-ended problems. As shown in Fig. C, the prediction probability rises from around layer 20, similar to the trend observed in multi-choice question answering problems.

### B.2 MULTIPLE TOKEN GENERATION

Building on our previous finding that the last token serves as the key checkpoint for video-language integration in open-ended generation, we further examine how checkpoints emerge within responses and how information propagates through these checkpoints as the model continues to generate multiple tokens.

**Experimental setup.** We adopt the Temporal QA subset of VCGBench (Maaz et al., 2024b), a video conversation benchmark that includes diverse reasoning-based QA examples. Specifically, we analyze how information flows among the video, question, generated response, and the last position, as the model continues to generate multiple temporal vocabularies. While automatically identifying temporal vocabulary in generated responses is challenging, verbs serve as strong candidates, since they often contain action and time-related semantics crucial for solving VideoQA tasks. To extract temporal vocabulary from model responses, we employed spaCy's en-core-web-lg model to detect verbs and used their token positions as semantic anchors for our analysis.

Then, we trace the probability change after Attention Knockout at each stage of anchor generation. For instance, given the question *"What is happening in this video?"* with baseline response *"A boy swings a bat and runs to the bases..."*, the detected anchors are [swings, runs, ...]. We then analyze information flow at different generation stages: generating the first anchor with no prior context (e.g., prompt: *"USER: <video> What is happening in this video? ASSISTANT: A boy"*, target: *"swings"*), generating the second anchor after one (e.g., prompt: *"USER: <video> What is happening in this video? ASSISTANT: A boy swings a bat and"*, target: *"runs"*), and so on.

**Results.** Figure D depicts the information flow of different routes: *video → question → last* (a–b), *video → last* (c), and *video → non-last response → last* (d–e). Our results show that, as generation continues, newly produced temporal verbs in the response increasingly function as additional core checkpoints. This is evidenced by a clear shift in the dominant sources feeding the last position. As the number of temporal anchors increases, the contribution of response to last consistently grows, while the relative importance of video to last and question to last decreases (Fig. D(a–c)). We also observe a structural change in the route through which video evidence reaches the final prediction. At the initial anchor generation with no given anchor, the model relies more on video to question to last (Fig. D(a–b)), whereas with a larger number of anchors it increasingly depends on video to the response to last (Fig. D(d–e)).

We observe consistent monotonic trends as the number of generated anchors increases from 1 to 3; therefore, we expect similar patterns to hold for more anchors. Overall, open-ended generation exhibits the same checkpoint-driven information flow pattern observed in multi-choice QA. The model dynamically forms new checkpoints around temporal concepts, and effective information flow reorganizes accordingly, confirming that our core claims generalize to open-ended VideoQA.

## C ADDITIONAL VISUALIZATION OF VIDEO-TO-QUESTION ATTENTION MAPS

We extend our analysis of emergent video concepts propagated through text tokens (Fig. 6) to various VideoQA tasks to verify the generalizability of our findings. To this end, we further conduct a qualitative analysis of video-to-question attention maps on the other tasks in TVBench.

As shown in Fig. E, the baseline model consistently exhibits attention on video tokens that are semantically aligned with the highlighted query words such as *down*, *first*, *from*, and *left*. For example, in the Action Antonym (Fig. E(a)) and Action Sequence tasks (Fig. E(b)), the query tokens focus on frames around the critical action change, while in the Scene Transition (Fig. E(c)) and Moving Direction (Fig. E(d)) tasks, they concentrate on frames that capture the transition of the scene or the motion of the object. In contrast, when cross-frame interactions are disabled, the question tokens fail to attend to relevant temporal vocabulary, resulting in indistinguishable attention maps across frames. This observation highlights the model's inability to capture temporal relationships and scene dynamics without cross-frame interaction, which supports the conclusion that our findings are broadly applicable to VideoQA models beyond specific task categories.

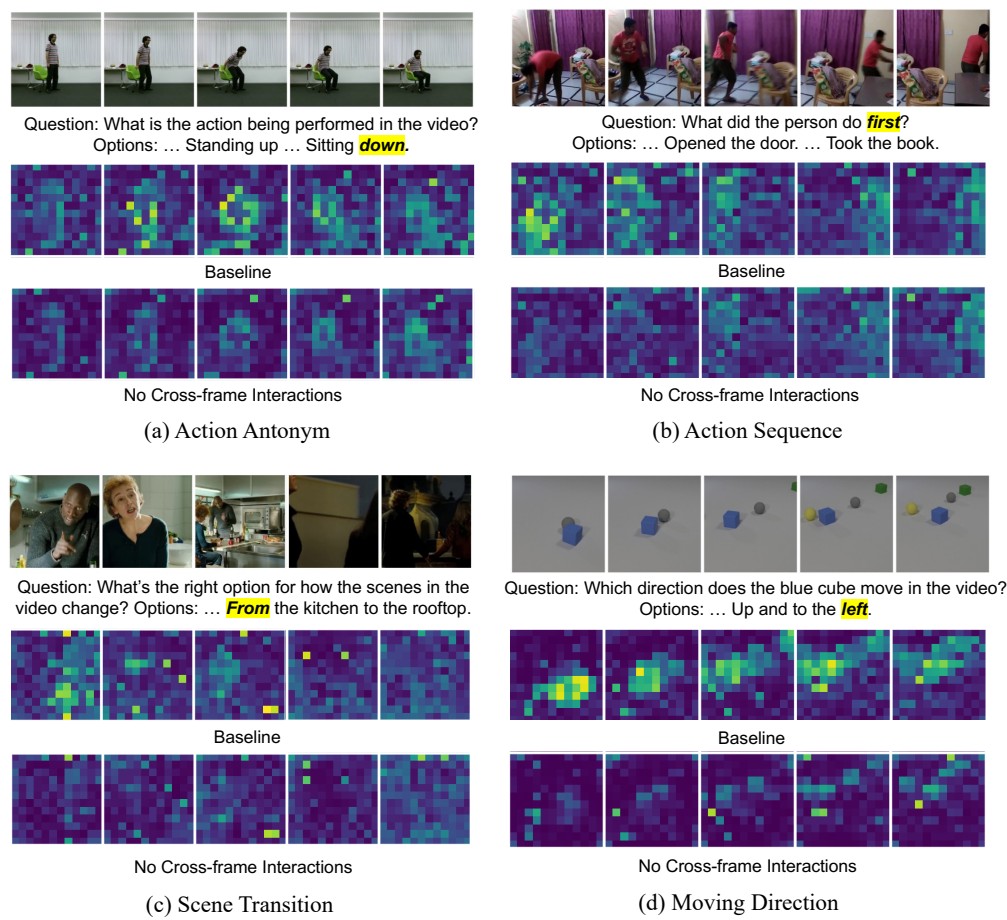

Figure E: **More visualizations of video-to-question attention maps.** Queries are highlighted in yellow, and keys correspond to video tokens. The baseline model attends to visually relevant tokens that align with the semantics of each query token, whereas disabling cross-frame interactions makes the model's attention less adaptive and limits its ability to infer the correct temporal context.

## D    FULL VISUALIZATION OF LOGIT LENS ANALYSIS

Figure F presents the full Logit Lens visualization referenced in Sec. 3.3. By visualizing the token positions corresponding to extracted spatial and temporal concepts, we observe a distinct ordering pattern: static concepts emerge in the earlier layers, while temporal concepts appear subsequently in deeper layers. Furthermore, the emergent positions of temporal concepts spatially align with their relevant foreground regions; for instance, the concept "sit" focuses on the region surrounding a seated person. Beyond this overall trend, spatial concepts tend to stabilize on salient regions early, and temporal concepts subsequently emerge primarily on the residual tokens rather than replacing already stabilized spatial tokens. We interpret this behavior as a consequence of priority in spatial localization, which explains the positioning mechanisms of temporal concepts.

## E    SCALABILITY OF OUR FINDINGS

To verify the scalability of our findings, we investigate a larger-scale VideoLLM. Specifically, we fine-tune LLaVA-Next-13B (Liu et al., 2024b) on video instruction tuning datasets, resulting in **LLaVA-NeXT-13B-Video-FT**. In this section, we analyze the information flow in the 13B model and show that the effective information flow pathways identified in the smaller model remain consistent at scale.

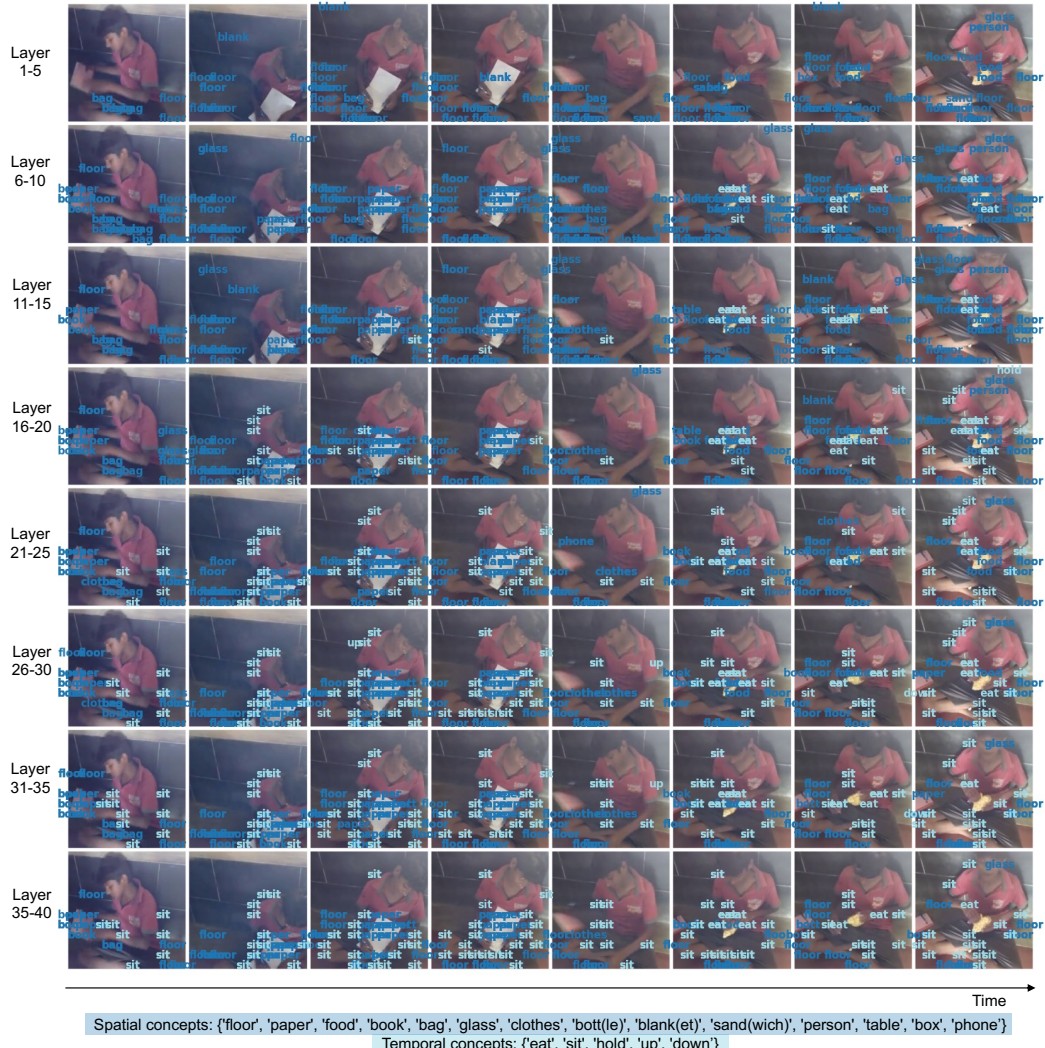

Spatial concepts: {'floor', 'paper', 'food', 'book', 'bag', 'glass', 'clothes', 'bott(le)', 'blank(et)', 'sand(wich)', 'person', 'table', 'box', 'phone'}
Temporal concepts: {'eat', 'sit', 'hold', 'up', 'down'}

Figure F: **Visualization of spatial and temporal concepts extracted by Logit Lens.** Spatial concepts tend to settle on salient regions early, and temporal concepts then emerge mainly on the remaining tokens rather than replacing already stabilized spatial tokens.

**Active temporal interaction within video tokens.** As shown in Fig. G, blocking cross-frame interactions in the early-to-middle layers consistently degrades the performance of the VideoLLM across all tasks (green), different from its ImageLLM baseline (pink). Notably, LLaVA-NeXT-13B-Video-FT exhibits similar trends to LLaVA-NeXT-7B-Video-FT, highlighting that our findings on active temporal interactions among video tokens generalize across model scales.

**Video-language integration on temporal keywords.** We further analyze the integration mechanism of video and text information in the larger-scale VideoLLM. Fig. H illustrates the impact of blocking attentions between video, question, and last position tokens. Consistent with the trends observed in the smaller-scale model, video information is not directly transmitted to the last token but is instead routed through the question tokens. To further trace how video and language information reaches the option tokens, we analyze the attention flow toward the answer options. As in Fig. J, information from video tokens flows to the true option tokens either directly or indirectly via non-option question tokens. The balance between these pathways varies across tasks, suggesting task-specific patterns in cross-modal integration. These consistent results showcase that our findings on video-language integration hold across scales.

Table A: **Impact of effective information flow pathways on LongVideoQA performance.** The total number of attention edges is calculated by counting valid (query, key) pairs over all attention layers.

| Case | Total Number of Attention Edges | Object-referred Event | Object before/ after Object | Scene-referred Object Tracking | All |
|---|---|---|---|---|---|
| Full causal attention | 25.7M (100%) | 52.9 | 40.9 | 44.4 | 46.1 |
| Attention in effective pathways | 10.8M (42%) | 54.0 | 39.4 | 43.2 | 45.5 |

**Answer generation.**  Figure K illustrates that generation occurs only after video and language information has been fused, primarily through the option tokens. The main difference from the smaller-scale VideoLLM lies in the specific layer range where this transition from integration to generation occurs, while the overall pattern remains consistent.

# F  GENERALIZATION OF OUR FINDINGS ACROSS VIDEOLLMS

In this section, we validate the generalization of our findings to other VideoLLMs. Specifically, we employ **VideoLLaMA3-7B** and **Mini-InternVL-4B-Video-FT**, a VideoLLM obtained by fine-tuning Mini-InternVL-4B (Gao et al., 2024) on video instruction tuning datasets. We verify VideoLLaMA3-7B and Mini-InternVL-4B-Video-FT to validate whether our findings on effective information flow generalize across different VideoLLMs.

**Active temporal interaction within video tokens.**  Fig. L and Fig. Q show that blocking the attention between video tokens leads to a greater performance drop across all tasks, indicating that Mini-InternVL-4B-Video-FT and VideoLLaMA3-7B also learn stronger temporal interaction through VideoQA training than its ImageLLM counterpart. Moreover, this behavior appears in the early-to-middle layers, similar to the behaviors of the LLaVA-NeXT series.

**Video-language integration on temporal keywords.**  We analyze how Mini-InternVL-4B-Video-FT and VideoLLaMA3-7B integrate video and language information in response to temporally grounded questions. As shown in Fig. M and Fig. R, video information is transmitted to question tokens in the early-to-middle layers, and only later transferred to the last position for answer generation. Fig. O and Fig. T further show that the video-language information is also gathered in the true option tokens, and the pathways toward the option tokens vary across the VideoQA tasks. These results demonstrate that our findings on video-language integration generally hold across various VideoLLMs.

**Answer generation.**  Fig. P and Fig. U show that although Mini-InternVL-4B-Video-FT tends to exhibit a sharp rise in generation probability across various VideoQA tasks, its overall behavior remains consistent with that of LLaVA-based VideoLLMs, where the probability begins to increase near the end of the video-language integration process.

# G  FURTHER ANALYSIS AND DISCUSSION

## G.1  QUANTITATIVE VALIDATION OF EFFECTIVE INFORMATION FLOW PATHWAYS ON LONG-FORM VIDEOS

We extend the effective pathway analysis to long video question-answering problems. To this end, we disabled the ineffective pathways of LLaVA-NeXT-7B-Video-FT as configured in Section 3.5 and evaluate the performance on LongVideoBench (Wu et al., 2024). Table A showcases that LLaVA-NeXT-7B-Video-FT also retains competitive performance on long-form videos understanding using only 42% of the original attention edges, with only a marginal accuracy drop of 0.6%p. This validates that our findings on the internal mechanisms of VideoLLMs generalize across various video question-answering tasks.

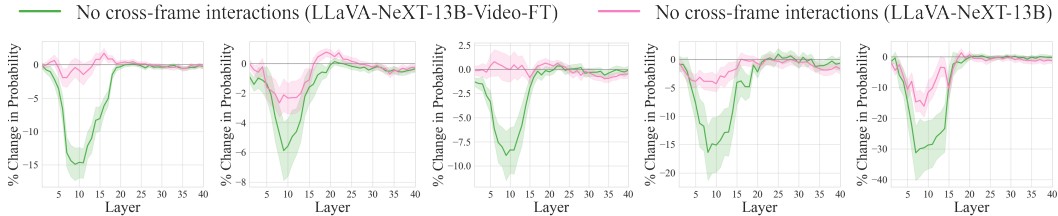

(a) Action Antonym  (b) Action Sequence  (c) Scene Transition  (d) Moving Direction  (e) Object Count

Figure G: **Change in prediction probability when disconnecting cross-frame attention edges in LLaVA-NeXT-13B-Video-FT and LLaVA-NeXT-13B.**

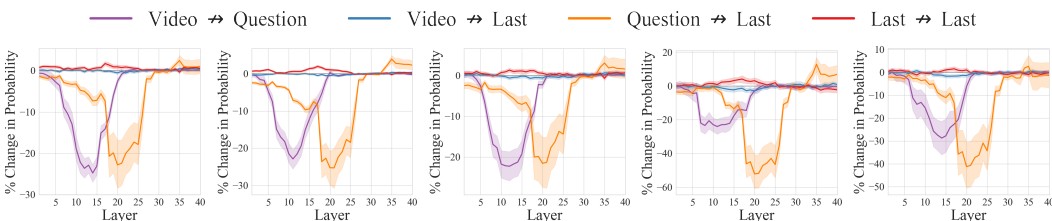

(a) Action Antonym  (b) Action Sequence  (c) Scene Transition  (d) Moving Direction  (e) Object Count

Figure H: **Change in the prediction probability of LLaVA-NeXT-13B-Video-FT when intervening on attention edges between video, question, and last token.**

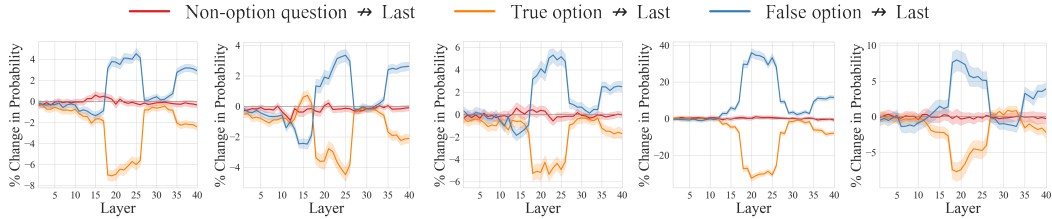

(a) Action Antonym  (b) Action Sequence  (c) Scene Transition  (d) Moving Direction  (e) Object Count

Figure I: **Change in the prediction probability of LLaVA-NeXT-13B-Video-FT when intervening on attention edges from different parts of the question tokens to the last token.**

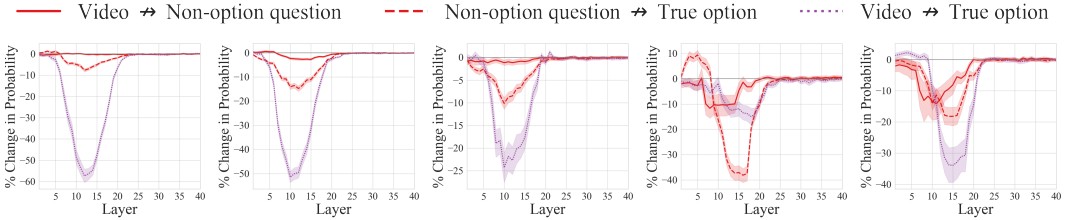

(a) Action Antonym  (b) Action Sequence  (c) Scene Transition  (d) Moving Direction  (e) Object Count

Figure J: **Change in the prediction probability of LLaVA-NeXT-13B-Video-FT when intervening on attention edges to the true option position.**

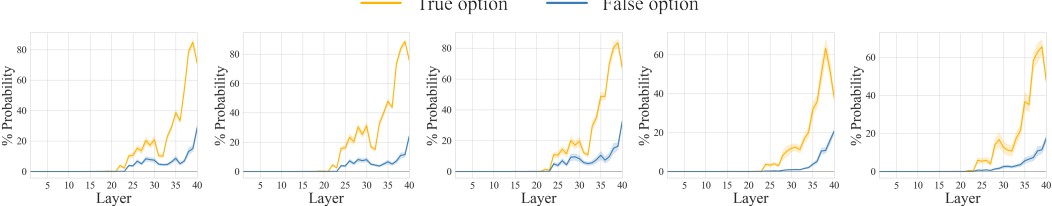

(a) Action Antonym  (b) Action Sequence  (c) Scene Transition  (d) Moving Direction  (e) Object Count

Figure K: **Layer-wise prediction probability of LLaVA-NeXT-13B-Video-FT for true and false options in the last token position.**

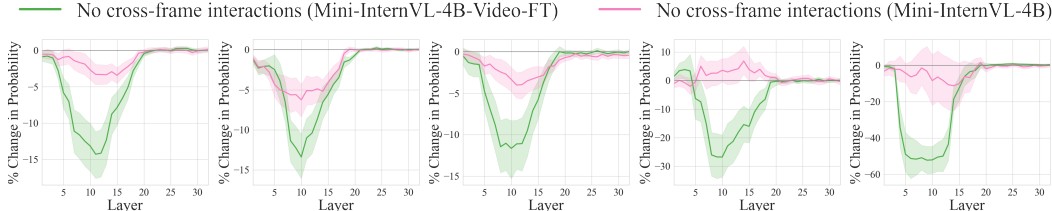

(a) Action Antonym  (b) Action Sequence  (c) Scene Transition  (d) Moving Direction  (e) Object Count

Figure L: **Change in prediction probability when disconnecting cross-frame attention edges in Mini-InternVL-4B-Video-FT and Mini-InternVL-4B.**

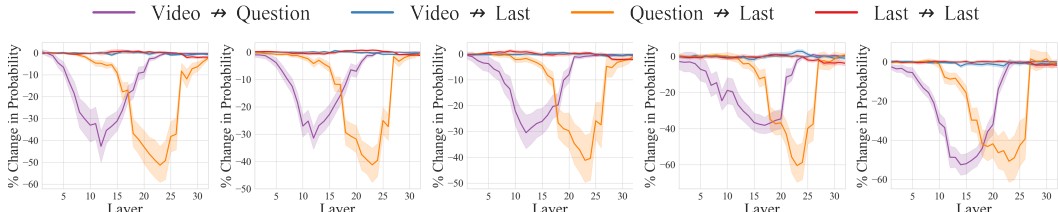

(a) Action Antonym  (b) Action Sequence  (c) Scene Transition  (d) Moving Direction  (e) Object Count

Figure M: **Change in the prediction probability of Mini-InternVL-4B-Video-FT when intervening on attention edges between video, question, and last token.**

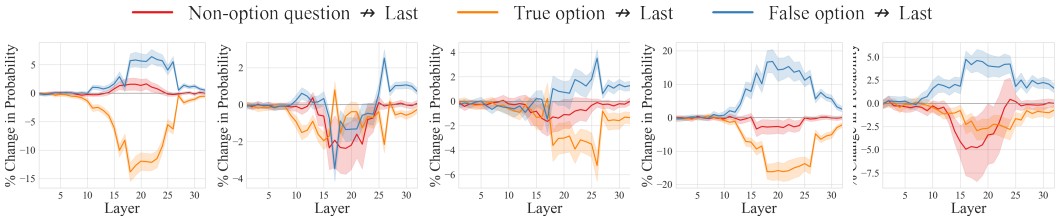

(a) Action Antonym  (b) Action Sequence  (c) Scene Transition  (d) Moving Direction  (e) Object Count

Figure N: **Change in the prediction probability of Mini-InternVL-4B-Video-FT when intervening on attention edges from different parts of the question tokens to the last token.**

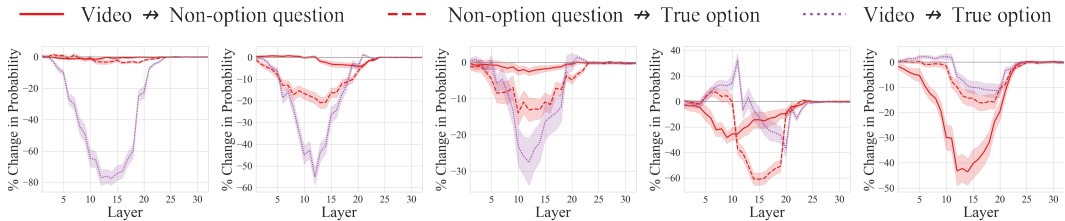

(a) Action Antonym  (b) Action Sequence  (c) Scene Transition  (d) Moving Direction  (e) Object Count

Figure O: **Change in the prediction probability of Mini-InternVL-4B-Video-FT when intervening on attention edges to the true option position.**

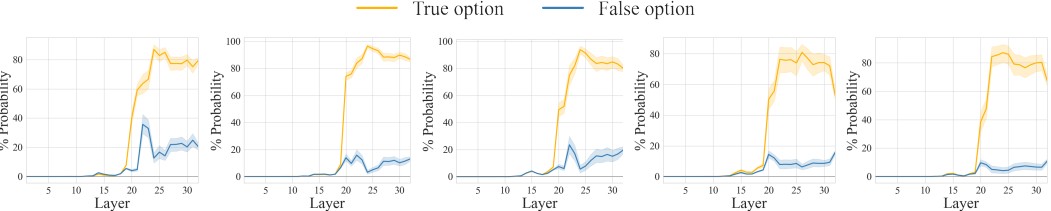

(a) Action Antonym  (b) Action Sequence  (c) Scene Transition  (d) Moving Direction  (e) Object Count

Figure P: **Layer-wise prediction probability of Mini-InternVL-4B-Video-FT for true and false options in the last token position.**

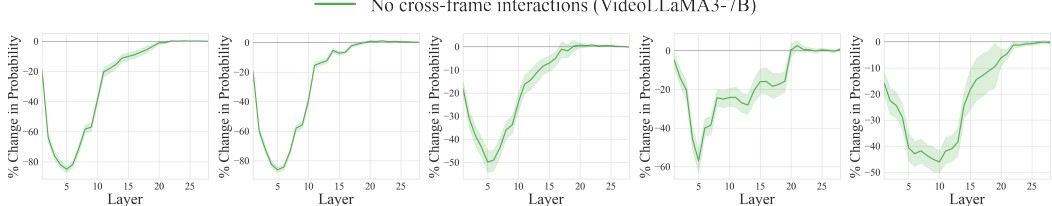

(a) Action Antonym  (b) Action Sequence  (c) Scene Transition  (d) Moving Direction  (e) Object Count

Figure Q: **Change in prediction probability when disconnecting cross-frame attention edges in VideoLLaMA3-7B.**

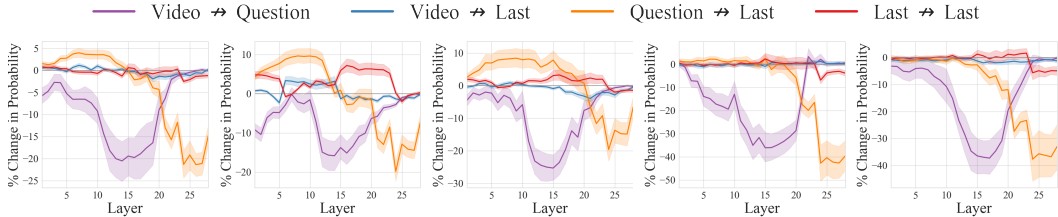

(a) Action Antonym  (b) Action Sequence  (c) Scene Transition  (d) Moving Direction  (e) Object Count

Figure R: **Change in the prediction probability of VideoLLaMA3-7B when intervening on attention edges between video, question, and last token.**

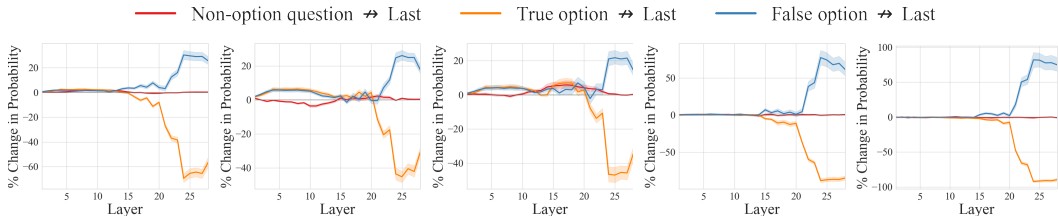

(a) Action Antonym  (b) Action Sequence  (c) Scene Transition  (d) Moving Direction  (e) Object Count

Figure S: **Change in the prediction probability of VideoLLaMA3-7B when intervening on attention edges from different parts of the question tokens to the last token.**

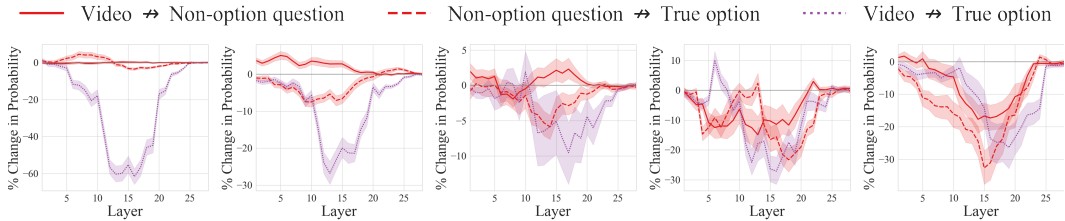

(a) Action Antonym  (b) Action Sequence  (c) Scene Transition  (d) Moving Direction  (e) Object Count

Figure T: **Change in the prediction probability of VideoLLaMA3-7B when intervening on attention edges to the true option position.**

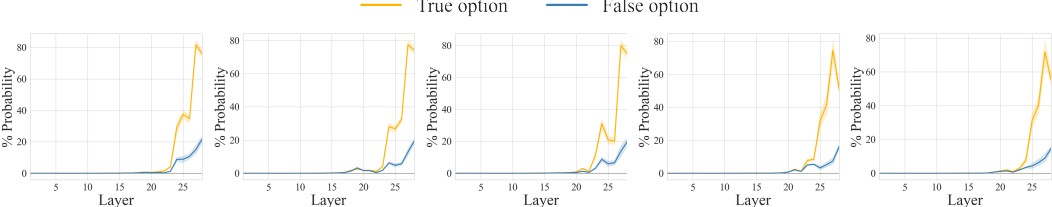

(a) Action Antonym  (b) Action Sequence  (c) Scene Transition  (d) Moving Direction  (e) Object Count

Figure U: **Layer-wise prediction probability of VideoLLaMA3-7B for true and false options in the last token position.**

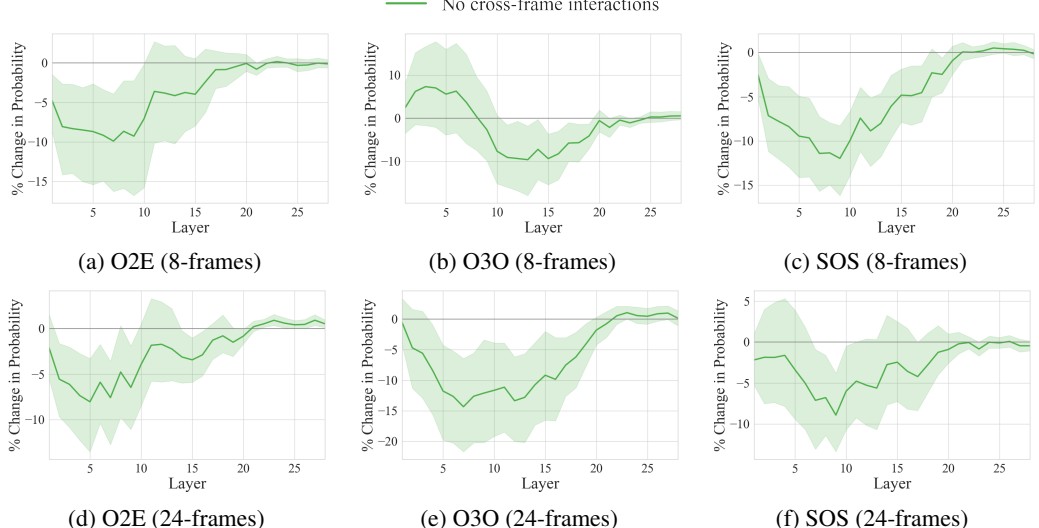

Figure V: **LongVideoBench: Changes in prediction probability after blocking cross-frame attention in VideoLLaMA3-7B with 8-frames (a–c) and 24-frames (d–f) inputs.** Object-referred Event (O2E), Object before/after Object (O3O), and Scene-referred Object Tracking (SOS) are used.

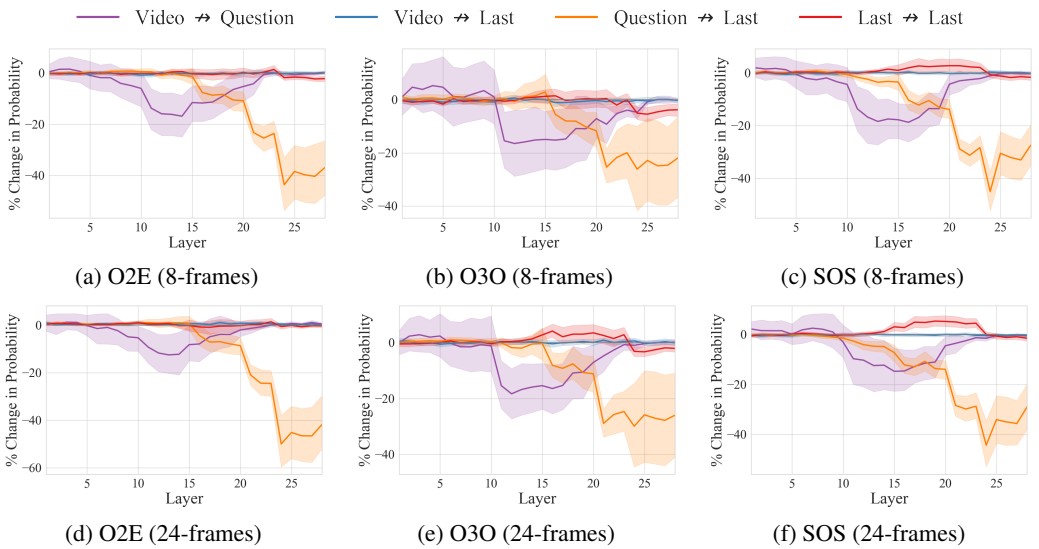

Figure W: **LongVideoBench: Changes in prediction probability when blocking attention edges between video, question, and last token with 8-frames (a–c) and 24-frames (d–f) inputs.**

## G.2 GENERALIZATION OF OUR ANALYSIS TO VARIOUS BENCHMARKS

We further investigate how the information flow changes with different forms of input videos and question types, including long video understanding in LongVideoBench (Wu et al., 2024) and spatial understanding in Video-MME (Fu et al., 2024).

**Long Video Understanding.** We validate VideoLLaMA3-7B on LongVideoBench (Wu et al., 2024) using the same setup in the main paper. Figure V, W, X, and Y highlight the results with 8-frame and 24-frame inputs, with $12 \times 12$ tokens per frame. Overall, the effective layer ranges for long-form VideoQA maintain similar patterns to short video benchmarks, regardless of the input frame lengths. A notable difference is that the probability drop from cross-frame attention and

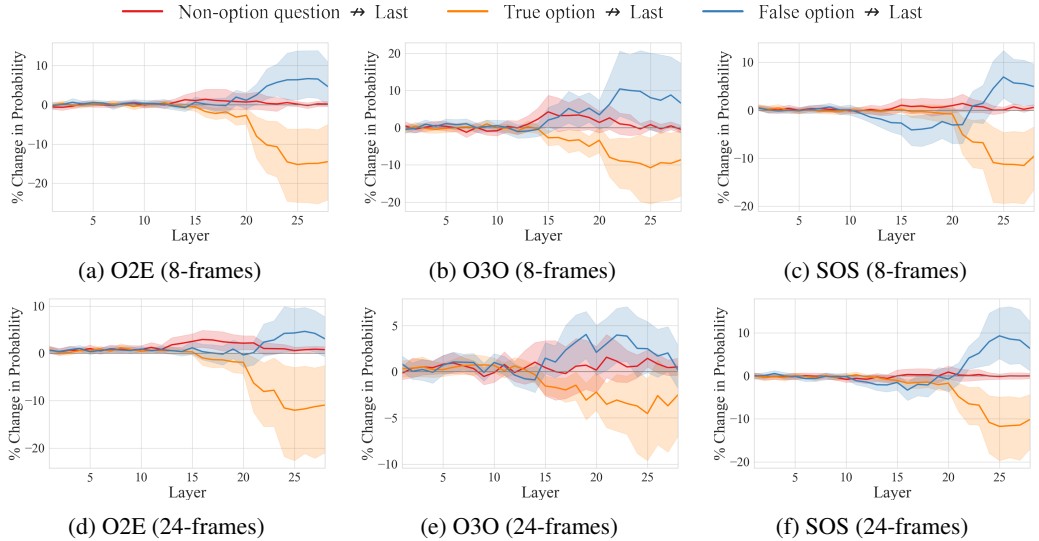

Figure X: **LongVideoBench: Prediction probability change when blocking attention edges from question tokens to the last token with 8-frames (a–c) and 24-frames (d–f) inputs.** Object-referred Event (O2E), Object before/after Object (O3O), and Scene-referred Object Tracking (SOS) are used.

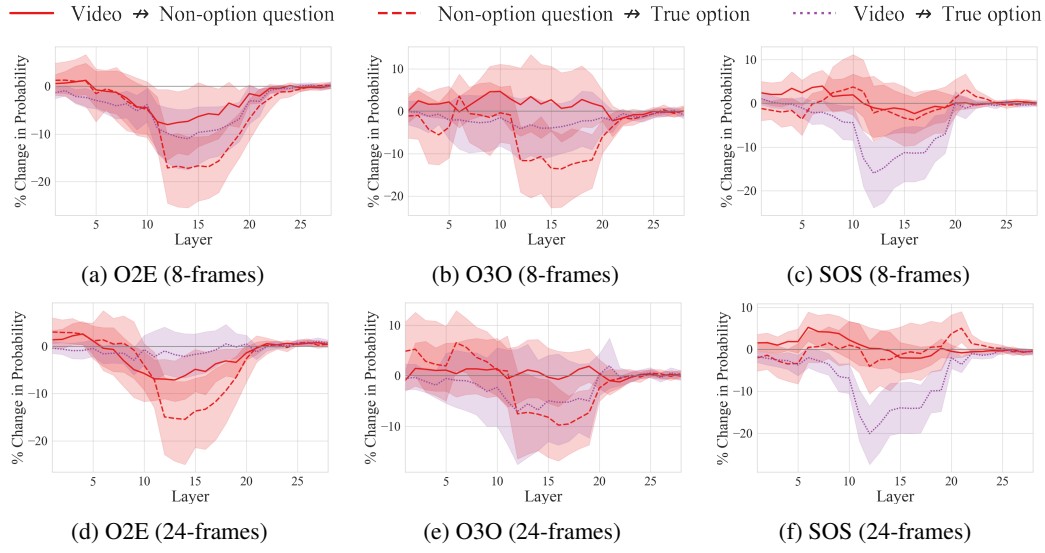

Figure Y: **LongVideoBench: Changes in prediction probability when blocking attention edges to the true option position in VideoLLaMA3-7B with 8-frames (a–c) and 24-frames (d–f) inputs.**

video-to-question components is relatively smaller compared to short video tasks (See Fig. Q and Fig. R). We conjecture this to two factors: (1) long video benchmarks typically do not require every frame to be equally informative, reducing the need for comprehensive visual processing, and (2) questions in long video benchmarks contain more descriptive information, causing the model to rely more heavily on textual cues from the question rather than visual content.

**Spatial Understanding.** To compare the patterns driven from spatial and temporal reasoning tasks, we adopt action recognition and spatial perception tasks from Video-MME (Fu et al., 2024). Results with LLaVA-NeXT-7B-Video-FT are shown in Fig. Z. When we block the cross-frame interaction, the action recognition task exhibits a clear and consistent performance drop, indicating that temporal aggregation is crucial for this setting. In contrast, the spatial perception task shows a much smaller

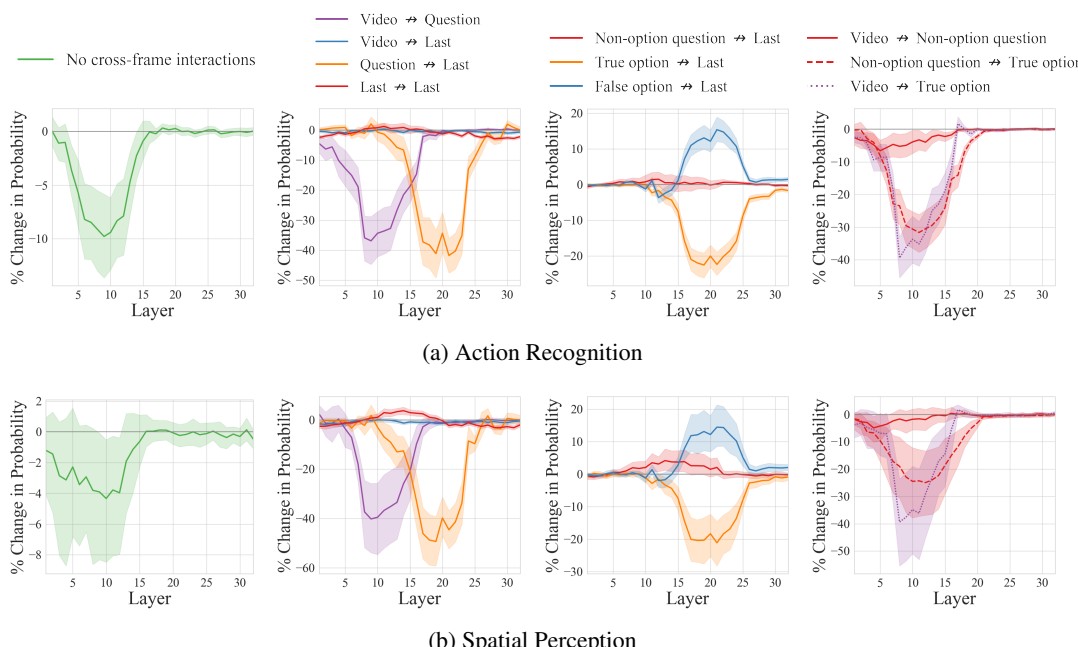

Figure Z: **Video-MME: Change in prediction probability when disabling each part of attention edges in LLaVA-NeXT-7B-Video-FT.** When cross-frame interactions are blocked, the spatial perception task shows a much smaller drop, as it contains questions that can be answered with static scenes. Other cross-modal information flow remains similar for both tasks.

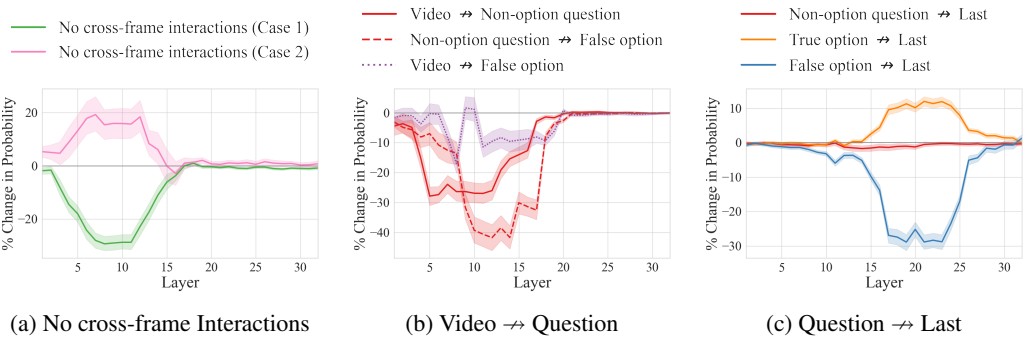

Figure AA: **Information flow in failed QA samples of Moving Direction task in TVBench.** Tracing the probability of the wrong answer in failed samples shows that errors stem from erroneous cross-frame attention (Case 1) or static bias (Case 2) during the early stage, rather than from a collapse of the cross-modal integration pathways.

average degradation and a much larger variance across samples. We conjecture that this pattern arises because some spatial perception questions incidentally benefits from temporal information, whereas many others can be answered from static scenes. Therefore, the impact of blocking cross-frame interaction ranges from almost no change to a significant drop.

## G.3   FAILURE CASE ANALYSIS

Continuing from Section 4, we have also conducted a failure case analysis in Moving Direction task of TVBench. Specifically, we trace how the probability of a wrong answer changes in the failure samples of LLaVA-NeXT-7B-Video-FT. As shown in Fig. AA (a), blocking cross-frame interactions shows two distinct patterns: (1) the model is highly confident in false options due to erroneous signals from failed cross-frame attention (green), or (2) the model relies on per-frame static scene information

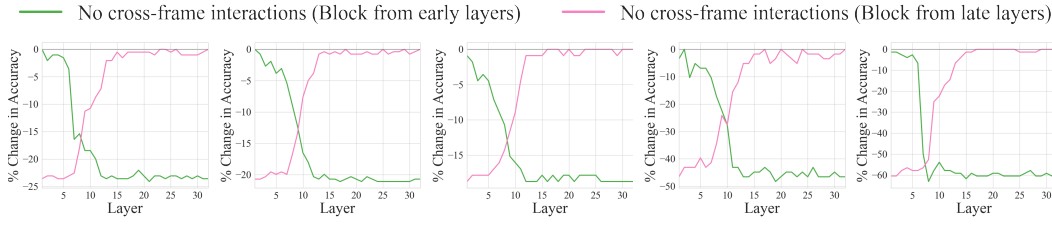

(a) Action Antonym  (b) Action Sequence  (c) Scene Transition  (d) Moving Direction  (e) Object Count

Figure AB: **Change in accuracy when gradually disconnecting cross-frame attention edges in each layer.** The green line shows the accuracy change when gradually blocking cross-frame attention from the first layer up to the $l^{th}$ layer, whereas the pink line shows the accuracy change when blocking from the $l^{th}$ layer to the last layer.

Table B: **Impact of cross-frame attention in the second half layers of VideoLLMs on answer generation.** We block cross-frame attention in the first and second half of the total layers and measure the resulting accuracy drop (%). While disabling cross-frame attention in the first half layers significantly degrades accuracy, disabling it in the second half layers barely impact performance.

| Case | Action Antonym | Action Sequence | Scene Transition | Moving Direction | Object Count |
|---|---|---|---|---|---|
| First half layers | −24.1 | −20.2 | −18.0 | −44.8 | −60.8 |
| Second half layers | −0.5 | −0.7 | −0.8 | −1.7 | −1.2 |

(pink). The cross-modal information flow shown in Fig. AA (b-c) remains similar to the successful samples, which implies that the failure mainly stems from the earlier spatiotemporal building process.

## G.4 SIGNAL LEAKAGE CHECK

Our Attention Knockout setup blocks cross-frame interaction within a local layer window, which may not capture potential residual signal leakage through bypassing pathways. To address this concern, we extend the intervention in two complementary ways. Instead of blocking cross-frame interaction only in the first half of the layers, we progressively expand the knockout from the first layer up to the $N$-th layer for $N = 1, \ldots, 32$. We also conduct the reverse intervention by blocking from the last layer backward. This gradual design explicitly check whether auxiliary signals from video tokens continue to propagate beyond early-to-middle layers, which would contradict the effective information flow range identified in our analysis. As shown in Fig. AB, progressively expanding the knockout does not lead to any additional significant performance drop unless the intervention overlaps with early-to-middle layers, the effective cross-frame interaction range that we identified. These results indicate that the cross-frame interaction actively occurs in the early-to-middle layer, supporting the validity of our findings beyond a specific knockout configuration.

## G.5 IMPACT OF BLOCKING CROSS-FRAME ATTENTION IN THE SECOND HALF LAYERS

We further examine the impact of blocking cross-frame attention in the second half layers. In Table B, the accuracy drop is marginal when temporal interactions are blocked in the latter layers, compared to the earlier layers. This supports our claim that active temporal interaction within video tokens occurs in early-to-middle layers.

## G.6 ROBUSTNESS OF OUR ANALYSES ON THE CHOICE OF WINDOW SIZE $k$

We observed the robustness of our analyses when using window sizes with sufficient width, and therefore chose to follow the $k$ value of 9 as used in (Geva et al., 2023). Specifically, to examine the robustness of our analyses to the window sizes, we conducted an extended analyses by varying the window sizes in 1, 5, 9, 13. As shown in Table AC, when the window size is extremely small (e.g., $k$=1), the narrow attention block is easily bypassed and VideoLLMs can still transmit information through remaining effective information pathways. This leads to only marginal probability drops

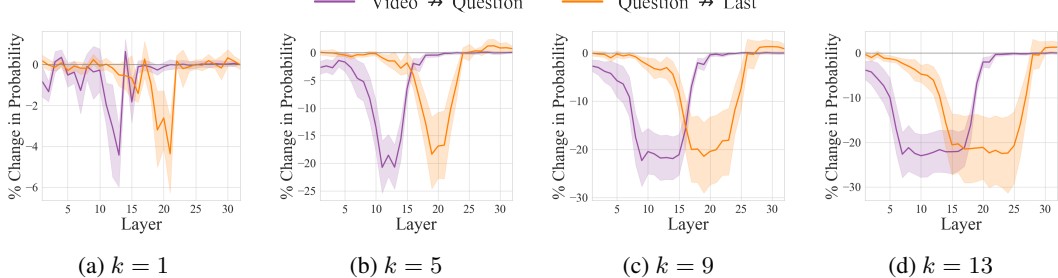

(a) $k = 1$        (b) $k = 5$        (c) $k = 9$        (d) $k = 13$

Figure AC: **Impact of window size $k$ on Attention Knockout.** Following Geva et al. (2023), we take $k = 9$ as our default choice.

Table C: **Coverage of models.** We adopt MLLMs with diverse sizes, base vision encoders, and base LLMs to ensure generalizable experimental validations.

| Model | Size | Base Vision Encoder | Base LLM |
|---|---|---|---|
| Mini-InternVL-4B | 4B | InternViT-300M-448px | Phi-3-mini-128k-instruct |
| LLaVA-NeXT-7B | 7B | CLIP-ViT-L-336px | Vicuna-7B-v1.5 |
| LLaVA-NeXT-13B | 13B | CLIP-ViT-L-336px | Vicuna-13B-v1.5 |
| VideoLLaMA3-7B | 7B | siglip-so400m-patch14-384 | Qwen2.5-7B |

across the layers. In contrast, with wider windows ($k$=5,9,13), we observe significant probability drops, which validates the robustness of our analyses across various choice of window sizes.

### G.7   DISCUSSION ON THE PROBABILITY INCREASES IN THE LAST LAYERS

We observe an increase in the true option probability in some cases when attention from the question to the last position is knocked out in later layers (e.g., Fig. 1(b)). Interleaving the attention knockout analysis from question to last (Fig. 3 and H) and the layer-wise prediction probability analysis (Fig. 9 and K) suggests that propagating information from question tokens to the last token boosts the true option, as this flow consolidates evidence in the middle layers. Therefore, the belief of VideoLLMs is already stabilized, so keeping this pathway open mainly acts as a broad amplifier, increasing probabilities for both true and false options. Thus, blocking the pathway only at the final layers preserves earlier propagated evidence for the true option in the hidden states while preventing further amplification of false options.

## H   IMPLEMENTATION DETAILS

We describe the implementation details for the VideoLLMs and their training setup. Table C shows the details of the VideoLLMs.

**Training setup.**   Our video instruction tuning data is derived from VideoChat2-IT (Li et al., 2024a), comprising 874k samples covering tasks such as VideoQA, captioning, reasoning, classification, and conversation. These samples are from diverse video understanding benchmarks, including VideoChatGPT-100k (Maaz et al., 2024b), VideoChat-11k (Li et al., 2023b), Webvid (Bain et al., 2021), YouCook2 (Zhou et al., 2018), TextVR (Wu et al., 2025), NExT-QA (Xiao et al., 2021), CLEVRER (Yi et al., 2019), TGIF (Li et al., 2016), Ego4D (Grauman et al., 2022), Kinetics-710 (Kay et al., 2017), and Something Something V2 (Goyal et al., 2017). We freeze the vision encoder while fully fine-tuning the MLP projector and LLM backbone. Our experiments are conducted with NVIDIA A6000 GPUs.

- **LLaVA-NeXT-7B-Video-FT.** During training, we initialize the model with LLaVA-NeXT-7B (Liu et al., 2024b), which employs CLIP-ViT-L-336px (Radford et al., 2021) as the vision encoder and Vicuna-7B-v1.5 (Zheng et al., 2023) as the language model. We utilize a batch size of 128 and train for 3 epochs. The base learning rate is initially set to 2e-5 and

Table D: **List of vocabularies used for semantic concept extraction.** Keywords are parsed from Action Sequence question prompts and converted to lowercase and present tense to avoid interference from linguistic completion in later layers.

| Spatial Keywords | bag, bed, blank, book, box, cabinet, camera, clothes, cup, door, floor, food, glass, laptop, paper, person, phone, sandwich, table |
|---|---|
| Temporal Keywords | close, down, drink, eat, hold, on, open, put, sit, take, throw, tidy, up |

Table E: **Effective pathway layer ranges for different VideoLLMs.** (a) Layer ranges for effective pathways across different models, determined by selecting 5-layer intervals with significant probability drops from Attention Knockout analysis. (b) Detailed knockout results showing probability drops across layer intervals in Action Antonym task. Significant drops ($< -5\%$) are highlighted in gray; N/A indicates unavailable layers.

(a) Effective pathway layer ranges

| Model | Cross-frame Interactions | Video-to-Question | Question-to-Last |
|---|---|---|---|
| LLaVA-NeXT-7B-Video-FT | L6-15 | L6-20 | L16-25 |
| LLaVA-NeXT-13B-Video-FT | L6-15 | L6-20 | L16-30 |
| Mini-InternVL-4B-Video-FT | L6-15 | L6-20 | L11-30 |
| VideoLLaMA3-7B | L1-15 | L6-20 | L21-28 |

(b) Attention Knockout results in Action Antonym

| Model | Interaction | L1-5 | L6-10 | L11-15 | L16-20 | L21-25 | L26-30 | L31-35 | L36-40 |
|---|---|---|---|---|---|---|---|---|---|
| LLaVA-NeXT-7B-Video-FT | Cross-frame | -4.2 | -11.1 | -6.3 | -0.2 | 0 | -0.2 | -0.2 | N/A |
| | Video-to-Question | -3.9 | -15.1 | -21.5 | -5.6 | -0.2 | 0 | 0 | N/A |
| | Question-to-Last | -0.3 | -1.2 | -4.5 | -19.3 | -15.1 | 0.7 | 1.1 | N/A |
| LLaVA-NeXT-13B-Video-FT | Cross-frame | -0.7 | -11.2 | -11.1 | -2.1 | -0.2 | -0.2 | -0.2 | -0.3 |
| | Video-to-Question | -1.2 | -16.7 | -29.1 | -9.2 | -0.3 | -0.2 | -0.2 | -0.2 |
| | Question-to-Last | -1.8 | -2.0 | -4.6 | -21.7 | -28.4 | -5.7 | -0.1 | -1.9 |
| Mini-InternVL-4B-Video-FT | Cross-frame | -2.3 | -11.1 | -11.5 | -3.3 | 0 | 0.2 | 0 | N/A |
| | Video-to-Question | -2.4 | -24.4 | -35.0 | -15.9 | -1.3 | -0.3 | -0.2 | N/A |
| | Question-to-Last | 0 | -1.3 | -5.9 | -30.8 | -46.8 | -14.1 | -3.2 | N/A |
| VideoLLaMA3-7B | Cross-frame | -65.1 | -61.4 | -14.9 | -5.0 | 0 | 0.2 | N/A | N/A |
| | Video-to-Question | -4.3 | -7.2 | -18.5 | -16.3 | -2.2 | 0 | N/A | N/A |
| | Question-to-Last | 2.0 | 3.7 | 1.9 | -2.7 | -16.2 | -19.1 | N/A | N/A |

is decayed to 5e-6 using a cosine scheduler, with a warmup ratio of 0.2. For both training and inference, we uniformly sample 8 frames as input and resize each frame into $336\times336$ pixels. These frames are then processed through a vision encoder to extract $8\times24\times24$ patch embeddings. Next, we use an MLP projector to project these embeddings, followed by average spatial pooling to generate $8\times12\times12$ video tokens.

- **LLaVA-NeXT-13B-Video-FT.** Similarly, we initialize the model with LLaVA-NeXT-13B (Liu et al., 2024b), which utilizes CLIP-ViT-L-336px (Radford et al., 2021) as the vision encoder and Vicuna-13B-v1.5 (Zheng et al., 2023) for the language model component. The model is trained for 1 epoch using the same training recipe and video token sampling strategy as LLaVA-NeXT-7B-Video-FT.

- **Mini-InternVL-4B-Video-FT.** We start with Mini-InternVL-4B (Gao et al., 2024), which adopts InternViT-300M-448px as the vision encoder and Phi-3-mini (Abdin et al., 2024) as the LLM backbone. We use a batch size of 128 with a learning rate of 4e-5, which decays to zero following a cosine schedule with a warmup ratio of 0.03 for a total of 3 epoch. For both training and inference, we uniformly sample 8 frames as input and resize each frame into $448\times448$ pixels. These frames are passed through the vision encoder, producing $8\times32\times32$ patch embeddings. After applying the MLP projection, we put $8\times16\times16$ video tokens as the input of the language model.

- **VideoLLaMA3-7B.** VideoLLaMA3-7B (Zhang et al., 2025a) uses SigLIP (Zhai et al., 2023) as a vision encoder and Qwen2.5-7B (Yang et al., 2024) as a LLM backbone. We directly use VideoLLaMA3-7B without fine-tuning and put $8{\times}12{\times}12$ video tokens as the input.

**Implementation details for Attention Knockout.** In VideoQA, a model generates an answer $a$ from a given video-question pair $(v, q)$, where the question may contain $n$ number of options $o = [o_1; o_2; ...; o_n]$. We employ Attention Knockout (Geva et al., 2023) to measure the information flow between different input parts. Specifically, the model initially predicts the answer $a$ with the highest probability $p_{\text{base}}$ at the last token position of the input sequence. After applying Attention Knockout as explained in § 2.2, we trace the relative change in probability $\%p_{\text{change}} = ((p_{\text{knockout}} - p_{\text{base}})/p_{\text{base}}) \times 100$, where $p_{\text{knockout}}$ is the updated probability for the same answer $a$ derived after intervention. Unless otherwise stated, we apply Attention Knockout within a window size of $k = 9$ layers around the $l^{\text{th}}$ layer of MLLMs, and trace the probability change for the first tokenized subword of the complete answer.

**Implementation details for Logit Lens.** To quantify emergence of spatial and temporal concepts in video tokens, we employ Logit Lens (nostalgebraist, 2020). We trace top-1 logits by projecting intermediate representations of all video tokens across layers using the language model head. We use Action Sequence videos with LLaVA-NeXT-13B-Video-FT. For the vocabulary pool, we parse spatial and temporal keywords from Action Sequence question prompts (Table D). To trace initial concept emergence, we convert parsed words to lowercase present tense, as linguistic completion occurs in later layers and can impact the analysis.

**Implementation details for effective pathway analysis.** To identify effective pathways, we use Attention Knockout results from Action Antonym tasks (Table E). We divide layers into 5-layer intervals, calculate average probability drops, and select intervals with significant drops $(< -5\%)$ as effective layers. We then enable cross-frame interactions, *video $\rightarrow$ question*, and *question $\rightarrow$ last* flows only within these effective layers while disabling *video $\rightarrow$ last* and *last $\rightarrow$ last* connections across all layers. Additionally, flows to *video* and *question* tokens are blocked in late layers as these tokens are no longer needed (e.g., after layers 20 and 25 respectively in LLaVA-NeXT-7B-Video-FT).

# I THE USAGE OF LARGE LANGUAGE MODELS

In this work, LLMs were used only to polish manuscript clarity, fix grammatical errors, and enhance readability. Specifically, all initial writing was done by the authors, with LLMs used afterwards for sentence-level polishing in part of the manuscript. LLMs were not involved in research ideation and experimental design. All core contributions, methodologies, and findings are the result of the authors' original work.

