# OpenReview forum: "Map the Flow: Revealing Hidden Pathways of Information in VideoLLMs"
_ICLR.cc/2026/Conference — ICLR 2026 Poster_

### Official Review · Reviewer_5oZh · 2025-10-26

**Soundness:** 4
**Presentation:** 4
**Contribution:** 3
**Rating:** 8
**Confidence:** 4

**Summary:**

This paper conducts a mechanistic interpretability study to uncover how Video Large Language Models (VideoLLMs) internally process temporal information for video question answering (VideoQA). Using techniques such as Attention Knockout and Logit Lens, the authors trace how video and text signals propagate across model layers. They identify a consistent three-stage information flow pattern: (1) temporal reasoning begins with active cross-frame interactions among video tokens in early-to-middle layers; (2) video-language integration emerges in middle layers as video representations align with linguistic embeddings tied to temporal concepts; and (3) answer generation occurs in middle-to-late layers once integration is complete. Additionally, the authors show that retaining only the identified “effective information pathways” preserves VideoQA performance while suppressing up to 58% of attention edges in LLaVA-NeXT-7B-Video-FT, demonstrating that these pathways are sufficient for temporal reasoning. The study further validates that these patterns generalize across model sizes, architectures, and both multiple-choice and open-ended question settings.

**Strengths:**

- This paper tackles an important yet under-explored problem: how spatiotemporal information is internally processed and propagated across layers in VideoLLMs.
- The experimental design is rigorous and well thought out, making effective use of mechanistic interpretability techniques such as Attention Knockout, Logit Lens, and attention map visualizations.
- The study successfully identifies the “effective information pathways” that underpin temporal reasoning in VideoLLMs, providing valuable insights into their inner workings.
- The paper is clearly written, logically structured, and easy to follow.

**Weaknesses:**

- The paper does not clearly articulate how its findings can be leveraged to guide the future development or improvement of VideoLLMs, limiting the practical implications of the insights.
- Most analyses are conducted on a single benchmark (TVBench), which may constrain the generalizability of the conclusions. Incorporating additional video temporal reasoning benchmarks (e.g., [1,2,3,4,5]) would strengthen the robustness of the results.
- The attention map visualizations in Fig. 5 are only presented for a single task. Including examples from all five task types listed in Table 1 would provide a more comprehensive understanding of the observed behaviors.
- The paper overlooks several relevant studies on video temporal reasoning benchmarks [1,2,3,4,5] and related methodological advances [6,7].

[1] Video-MME: The First-Ever Comprehensive Evaluation Benchmark of Multi-modal LLMs in Video Analysis.

[2] TempCompass: Do Video LLMs Really Understand Videos?

[3] MotionBench: Benchmarking and Improving Fine-grained Video Motion Understanding for Vision Language Models.

[4] Vinoground: Scrutinizing LMMs over Dense Temporal Reasoning with Short Videos.

[5] TemporalBench: Benchmarking Fine-grained Temporal Understanding for Multimodal Video Models.

[6] Temporal Preference Optimization of Large Multimodal Models.

[7] Temporal Reasoning Transfer from Text to Video.

**Questions:**

- If we perform the same analysis on non-temporal video understanding tasks (e.g., "What is the main object presented in the video?"), what would the information flow be like?

---

> ### Author Response · Authors · 2025-11-22
> **Author Response to Reviewer 5oZh (1/3)**
>
> We sincerely thank Reviewer 5oZh for the thoughtful comments and constructive suggestions. We appreciate the positive feedback highlighting our paper’s strengths, including the importance of the problem, the rigor of our experimental design, successful identification of effective pathways and insights on inner workings, and clear writing.
>
> We first list up the modified part in the revised paper corresponding to each comment:
> * **W1. Future applications of our findings**
>     * "Future applications of our findings" paragraph in Section G
> * **W2. Extension to additional benchmarks**
>     * Section F.2 Generalizability of our analysis to various benchmarks and Figure X, Y, Z, AA, AB, AC, AD, AE, AF, AG, AH, and AI
> * **W3. Additional attention map visualization for all tasks**
>     * Section E "Additional visualization of video-to-question attention maps" and Figure U
> * **Q1. Investigation on spatial understanding VideoQA task**
>     * "Spatial understanding" paragraph in Section F.2 "Generalizability of our analysis to various benchmarks" and Figure AF, AG, AH, and AI
> Below, we provide detailed responses to each point.
>
>
>
> > ### **W1. Future applications of our findings**
>
> Following the advice, we further discuss the applicability of our observations for training and testing. For training, current VideoLLMs often use only a limited part of the model to propagate information. Therefore, we may expand the utility of the model for such propagation by intentionally blocking the currently dominant information pathways during training. This encourages the VideoLLM to utilize alternative routes for information propagation, which can further exploit the potential capacity of the model. For testing, our findings suggest that we can apply early-exiting [1,2,3] to tokens that lie beyond the range of effective information flow pathways, which may reduce the overall computation cost without significantly affecting accuracy.
>
>
> [1] Maha Elbayad, Jiatao Gu, Edouard Grave, and Michael Auli. 2020. Depth-adaptive transformer. In 8th International Conference on Learning Representations, ICLR 2020
>
> [2] Tal Schuster, Adam Fisch, Jai Gupta, Mostafa Dehghani, Dara Bahri, Vinh Tran, Yi Tay, and Donald Metzler. Confident adaptive language modeling. Advances in Neural Information Processing Systems, 2022
>
> [3] Sangmin Bae, Jongwoo Ko, Hwanjun Song, Se-Young Yun, Fast and Robust Early-Exiting Framework for Autoregressive Language Models with Synchronized Parallel Decoding, EMNLP 2023

---

> ### Author Response · Authors · 2025-11-22
> **Author Response to Reviewer 5oZh (2/3)**
>
> > ### **W2. Extension to additional benchmarks**
>
> Thank you for the recommendation. Following your advice, we have added Video-MME and LongVideoBench as additional benchmarks to ensure the generalizability of our findings on diverse video and question types. Overall, we confirm that our observations on effective information flow pathways still hold for diverse types, and only the dominance of cross-frame interactions on the final performance could differ.
>
> **Video-MME.** We extend our analysis to the action recognition and spatial perception tasks in the Video-MME benchmark. As shown in Fig. AF, AG, AH, and AI, the main patterns we report in the paper remain valid on these tasks. Video language integration is concentrated in the middle layers, and the answer options consistently act as core checkpoints that organize and guide the information flow. These consistent trends across benchmarks demonstrate that our findings generally hold for a wide range of VideoQA benchmarks. However, when we analyze cross-frame interaction on these tasks, we observe a clear difference between the two tasks. While our finding on cross-frame interaction holds in the action recognition task, the spatial perception task shows a much smaller average drop and a much larger variance. We hypothesize that many questions in this task can be solved from static scene information alone, which reduce its dependency on cross-frame interaction, while only a subset truly requires temporal information. This mixture of questions leads to diverse behaviors ranging from almost no degradation to substantial performance loss.
>
> **LongVideoBench.** Furthermore, we have extended our analysis to longer video setups. We validated VideoLLaMA3 on LongVideoBench, a video reasoning benchmark featuring complex, longer videos with tasks requiring sophisticated reasoning, such as event localization. We used Object-referred Event (O2E), Object before/after Object (O3O), and Scene-referred Object Tracking (SOS). We have compared 8 frames (Fig. X, Y, Z, and AA) and 24 frames (Fig. AB, AC, AD, and AE) with 12x12 tokens per frame. The effective layer ranges for long-form VideoQA maintain similar patterns to short video benchmarks. One notable difference is that the probability drop from cross-frame attention and video-to-question components is relatively smaller compared to short video benchmarks. We conjecture this to two factors: (1) long video benchmarks typically don't require every frame to be equally informative, reducing the need for comprehensive visual processing, and (2) questions in long video benchmarks contain more descriptive information, causing the model to rely more heavily on textual cues from the question rather than visual content. We confirm that our observations on effective information flow pathways still work even with longer and more complex video question-answering samples.
>
>
>
> > ### **W3. Additional attention map visualization for all tasks**
>
> We further visualize attention maps for representative samples from each task (i.e., action antonym, action sequence, scene transition, moving direction, object count). As shown in Fig. U, we consistently observe that temporal vocabulary tokens effectively trigger cross-frame interaction and then propagate the aggregated temporal information to the language tokens. Specifically, VideoLLM consistently distributes attention on video tokens that are semantically aligned with the highlighted query words such as **down**, **first**, **from**, and **left**. For example, in the action antonym (Fig.U(a)) and action sequence tasks (Fig. U(b)), the query tokens focus on frames around the critical action change, while in the scene transition (Fig.U(c)) and moving direction (Fig.U(d)) tasks they concentrate on frames that capture the transition of the scene or the motion of the object. However, when cross-frame interactions are disabled, the attention of the video tokens is not triggered by their relevant temporal vocabulary, showing undistinguishable attention maps across frames. These consistent patterns across diverse tasks validate the generalizability of our findings.
>
>
>
>
> > ### **W4. Recommendation for the missing relevant studies**
>
> We have reflected the recommended references on video temporal reasoning benchmarks and related methodological advances in "VideoQA Benchmark" and "Video Large Language Models (VideoLLMs)" in the related work section in the revised paper. Thank you for the recommendation.

---

> ### Author Response · Authors · 2025-11-22
> **Author Response to Reviewer 5oZh (3/3)**
>
> > ### **Q1. Investigation on spatial understanding VideoQA task**
>
> Following the comment, we extend our analysis to VideoQA for spatial understanding tasks. To this end, we investigate two tasks of the Video-MME benchmark: the action recognition task and the spatial perception task. As shown in Fig.  AF, when we block the cross-frame interaction, the action recognition task exhibits a clear and consistent performance drop, indicating that temporal aggregation is crucial for this setting. In contrast, the spatial perception task shows a much smaller average degradation and a much larger variance across samples. We conjecture that this pattern arises because some spatial perception questions incidentally benefit from temporal information, whereas many others can be answered from static scenes. Therefore, the impact of blocking cross-frame interaction ranges from almost no change to a significant drop. Overall, this analysis supports our claim that spatial understanding VideoQA tasks are less sensitive to cross-frame interaction.

---

> > ### Comment · Reviewer_5oZh · 2025-11-25
> > **Response to Author Response**
> >
> > I thank the authors for their comprehensive response to my comments, which has effectively address my earlier concerns and further strengthen the paper. Overall, I believe this work provides a solid contribution to understanding the inner mechanisms of Video LLMs, and I am pleased to maintain my positive evaluation.

---

> > > ### Author Response · Authors · 2025-11-25
> > > **Official Comment by Authors**
> > >
> > > Thank you so much for carefully reading and reviewing our response. We're glad that our revisions and response have addressed your concerns. Please feel free to let us know if there are any remaining points we can further clarify. Thank you again for your valuable comments!

---

### Official Review · Reviewer_trVM · 2025-11-01

**Soundness:** 3
**Presentation:** 3
**Contribution:** 3
**Rating:** 6
**Confidence:** 2

**Summary:**

This paper investigates the internal mechanisms of Video Large Language Models (VideoLLMs) on VideoQA tasks using mechanistic interpretability techniques. By applying tools such as Attention Knockout and Logit Lens, the authors identify a consistent three-stage information flow: (1) cross-frame interactions among video tokens in early-to-middle layers, (2) video-language integration onto temporal keywords in the middle layers, and (3) answer generation in the middle-to-late layers. The authors validate these "effective pathways" by demonstrating that a pruned model, which retains only these key connections (e.g., 42% of edges), largely maintains its baseline performance.

**Strengths:**

1. The paper addresses a critical and under-explored area. As VideoLLMs grow in capability, understanding how they perform temporal reasoning (their internal mechanism) is as important as if they can (their performance). Applying established interpretability techniques from text/images to the spatiotemporal domain is a valuable and logical contribution.
2. The paper's main contribution is a clear "blueprint" of the temporal reasoning process in VideoLLMs (Fig. 1). The identified 3-stage pathway—cross-frame interaction, integration, and generation—is intuitive and provides a strong conceptual framework for the research community.

**Weaknesses:**

1. Selection bias in analysis: The analysis is exclusively performed on samples where the model outputs the correct answer. This introduces a selection bias, as the study only explains the mechanism of successful reasoning. A complete mechanistic understanding should also include a failure-mode analysis. It is unclear if model errors are due to a breakdown in these same pathways (e.g., failed video-language integration) or if they follow entirely different, pathological pathways.
2. Limited Temporal Scope of Inputs: The entire analysis is conducted on models processing only 8 frames of video. This is a short temporal window, and the findings may not generalize to more complex, long-form video reasoning.
3. Reliance on Multiple-Choice Format: The paper's core findings—such as the convergence of information onto the "true option" tokens —may be an artifact of the multiple-choice task format itself. This format steers the model toward a "verification" task (i.e., selecting the best option) rather than a "generative" one. This makes it unclear if the identified pathway is fundamental to temporal reasoning or just a learned shortcut for solving MC-QA. While the Appendix (SC) attempts to address this with an open-ended analysis, this analysis is limited, as it only includes three of the five tasks, potentially omitting more complex reasoning scenarios.

**Questions:**

See weaknesses above.

---

> ### Author Response · Authors · 2025-11-22
> **Author Response to Reviewer trVM (1/2)**
>
> We sincerely thank Reviewer trVM for the thoughtful comments and constructive suggestions. We appreciate the positive feedback highlighting our paper’s strengths, including the importance of addressing this critical and under-explored area, the valuable contribution to understanding inner mechanisms in VideoLLMs, and the clear three-stage pathways providing an intuitive conceptual framework for the research community.
>
> We first list up the modified part in the revised paper corresponding to each comment:
> * **W1. Analysis on the failure VideoQA samples**
>     * Section F.3 "Failure case analysis" with Fig. AJ, AK, and AL
> * **W2. Further investigation on the information flow for complex reasoning & long-form videos**
>     * "Long video understanding" paragraph in Section F.2 and Fig. AB, AC, AD, and AE.
> * **W3. Further investigation on the information flow for open-ended questions**
>     * Section C.2 "Multiple Token Generation" and Figure S
>
> Below, we provide detailed responses to each point.
>
>
>
> > ### **W1. Analysis on the failure VideoQA samples**
>
> Thank you for the valuable comment. To offer a mechanistic understanding of why the model makes wrong answers, we have extended our analysis to failure cases. We focus on two aspects: (1) samples where the VideoLLM is highly confident in false options, and (2) samples when the VideoLLM highly relies on static scene information rather than temporal reasoning.
>
> To this end, we analyze how the probability of a wrong answer changes in the failure VideoQA samples. As can be seen in Fig. AK and Fig. AL, the cross-modal flow patterns routing false options are the same as those with successful samples routing true options, indicating that a root cause could be in an earlier stage of building video representations. In contrast, when we intervene on the cross-frame interaction as in Fig. AJ, failed cases split into two patterns. In some examples, the probability of the incorrect option decreases (green), while in others it increases (pink), and thus no single consistent behavior emerges.
>
> We conjecture that, in the first type, the VideoLLM is already overconfident in false options, where the erroneous signal may come from cross-frame interaction or misaligned between specific video and language. In the second type, a plausible interpretation is that the model primarily relied on per-frame static scene information. Thus, when cross-frame interaction is interrupted, the model even more emphasizes the rationale from static scenes, which in turn reinforces confidence in false options relevant to those static scenes.
>
>
> > ### **W2. Further investigation on the information flow for complex reasoning & long-form videos**
>
> Following the advice, we have extended our analysis to long-form videos. We have validated VideoLLaMA3 on LongVideoBench, a video reasoning benchmark featuring complex, longer videos with tasks requiring sophisticated reasoning such as event localization. We have adopted Object-referred Event (O2E), Object before/after Object (O3O), and Scene-referred Object Tracking (SOS). We used 24 frames with 12x12 tokens per frame. As shown in Fig. AB, AC, AD, and AE, the information flow pathways for long-form VideoQA maintain similar patterns to short video benchmarks (Fig. K, L, M, and N). A notable difference is that the probability drop from cross-frame attention and video-to-question components is relatively smaller compared to short video benchmarks. We conjecture this to two factors: (1) long video benchmarks typically don't require every frame to be equally informative, reducing the need for comprehensive visual processing, and (2) questions in long video benchmarks contain more descriptive information, causing the model to rely more heavily on textual cues from the question rather than visual content. We confirm that our observations on effective information flow pathways still work even with longer and more complex video question-answering samples.

---

> ### Author Response · Authors · 2025-11-22
> **Author Response to Reviewer trVM (2/2)**
>
> > ### **W3. Further investigation on the information flow for open-ended questions**
>
> We agree that extending our observations from multi-choice QA to open ended VideoQA is important for validating the scope of our claims. Therefore, beyond analyzing the isolated case of generating a single temporal vocabulary token in Section C.1, we further investigate how the information flow is affected as the response is generated progressively when solving open-ended questions.s. Building on our findings in Section C.1 that the last token serves as the key checkpoint for video-language integration, we analyze how information flows among the video, question, generated response, and the last position evolves as generation proceeds.
>
>
> **Experimental setup.** We adopt the Temporal QA subset of VCGBench, a video conversation benchmark that includes diverse reasoning-based QA examples with different question types. While automatically identifying temporal vocabulary in generated responses is challenging, verbs serve as strong candidates, since they often contain action and time-related semantics crucial for solving VideoQA tasks. To extract temporal vocabulary from model responses, we employed spaCy's en-core-web-lg model to detect verbs and used their token positions as semantic anchors for our analysis.
>
> Then, we trace the probability change after Attention Knockout at each stage of anchor generation. For instance, given the question "What is happening in this video?'' with baseline response "A boy swings a bat and runs to the bases...", the detected anchors are \[swings, runs, ...\]. We then analyze information flow at different generation stages: generating the first anchor with no prior context (e.g., prompt: "USER: \<video\> What is happening in this video? ASSISTANT: A boy", target: "swings"), generating the second anchor after seeing one anchor (e.g., prompt: "USER: \<video\> What is happening in this video? ASSISTANT: A boy swings a bat and", target: "runs"), and so on.
>
> **Results.** Figure S depicts the effective information flow of different routes: video → question → last (a-b),  video → last (c), and video → response → last (d-e), as we vary the number of anchors from 1 (i.e., the experimental setup in Section C.2) to 2 and 3. Our results show that, as generation continues, newly produced temporal verbs in the response increasingly function as additional core checkpoints. This is evidenced by a clear shift in the dominant sources feeding the last position. As the number of temporal anchors increases, the contribution of response to last consistently grows, while the relative importance of video to last and question to last decreases (Fig. S(a), (b), (c)). We also observe a structural change in the route through which video evidence reaches the final prediction. When the number of anchors is 1, the model relies more on video to question to last, whereas with a larger number of anchors, it increasingly depends on video to response to last (Fig. S(d), (e)). We observe consistent monotonic trends as the number of anchors increases from 1 to 3, and therefore expect similar patterns to hold for more anchors. Overall, open-ended generation exhibits the same checkpoint-driven reasoning pattern observed in multi-choice QA. The model dynamically forms new checkpoints around temporal verbs, and effective information flow reorganizes accordingly, confirming that our core claims generalize to open-ended VideoQA without fixed answer options.

---

> ### Author Response · Authors · 2025-11-28
> **Looking Forward to Your Feedback**
>
> Dear reviewer trVM,
>
> We sincerely appreciate the time and effort you’ve dedicated to reviewing our paper. As the discussion period ends soon on December 3rd, we wanted to kindly check if our responses have adequately addressed your concerns. If there are any remaining questions or points you'd like us to clarify, we would be more than happy to provide further details. On the other hand, if our rebuttal has resolved your concerns, we would be truly grateful if you could reflect that in your final evaluation.
>
> Thank you again for your thoughtful review, and we look forward to your feedback.
>
> Sincerely, The Authors

---

### Official Review · Reviewer_wzb7 · 2025-11-01

**Soundness:** 3
**Presentation:** 4
**Contribution:** 2
**Rating:** 8
**Confidence:** 3

**Summary:**

This paper focuses on uncovering the internal information flow mechanisms of VideoLLMs for VideoQA tasks. Using interpretability techniques (e.g., Attention Knockout), it analyzes how VideoLLMs extract and propagate spatiotemporal and textual information across layers and modalities. It identifies consistent temporal reasoning patterns across diverse VideoQA tasks and validates the sufficiency of "effective information pathways" for maintaining model performance.

**Strengths:**

Overall, this paper is well written.

1. This paper pioneers using mechanistic interpretability (Attention Knockout, Logit Lens) to unpack internal information flow, which is an interesting and novel topic.

2. Multiple models (LLaVA-NeXT-7B/13B, Mini-InternVL-4B, VideoLLaMA3-7B) and benchmarks (TVBench, TOMATO, LongVideoBench) are tested, with validations for scalability and generalization, ensuring reliability.

**Weaknesses:**

The paper observes that spatial concepts emerge in very early layers and temporal concepts in middle layers (via Logit Lens analysis on video tokens) but fails to probe the underlying causes. For instance, it lacks ablations on critical factors directly tied to this phenomenon: visual encoder design (e.g., CLIP-ViT-L-336px in LLaVA-NeXT models vs. SigLIP in VideoLLaMA3-7B, as noted in Table C) and the temporal information density of video fine-tuning data—both of which could explain why concept emergence timelines differ. Deeper investigation into these root factors would significantly strengthen the persuasiveness of its findings on concept emergence patterns.

**Questions:**

See weaknesses.

---

> ### Author Response · Authors · 2025-11-22
> **Author Response to Reviewer wzb7**
>
> We sincerely thank Reviewer wzb7 for the thoughtful comments and constructive suggestions. We appreciate the positive feedback stating that our paper is well written, features a novel and interesting topic of unpacking internal information flow, and is validated with multiple models and benchmarks, thereby ensuring scalability and generalizability.
>
> We supplement the explanation for the comment in Section D "Visualization of logit lens" and Figure T.
> Below, we provide a detailed response to your comment.
>
>
> > ### **W1. Deeper analysis on why the concept emergence timelines differ**
>
>
> Following the advice, we further investigate the underlying reason on concept emergence timelines differ. To this end, we visualize the token positions corresponding to spatial / temporal concepts using Logit Lens [1]. As shown in Fig. T, static concepts emerge first, and temporal concepts appear later as layers go deeper. Moreover, the emergent positions of temporal concepts spatially align with their relevant foreground regions, for example, the concept “sit” aligns with the region around a seated person. This result is consistent with our finding on concept emergence in video tokens in Section 3.3. Beyond this overall trend, we additionally observe that spatial concepts tend to settle on salient regions early, and temporal concepts then emerge mainly on the remaining tokens rather than replacing already stabilized spatial tokens. We interpret this behavior as a consequence of priority in spatial localization. Specifically, foreground regions are first mapped with spatial concepts that describe salient entities or attributes, and temporal concepts tend to emerge afterward. We hypothesize that this order of emergence enforces temporal concepts to occupy the remaining token positions not already taken by spatial concepts, which explains the positioning mechanisms of temporal concepts. Overall, this analysis provides a more concrete explanation of the emergence order of spatial and temporal concepts and their effect on where temporal concepts are grounded.
>
>
> [1] nostalgebraist. Interpreting GPT: The logit lens. https://www.lesswrong.com/posts/ AcKRB8wDpdaN6v6ru/interpreting-gpt-the-logit-lens, August 2020.

---

> ### Author Response · Authors · 2025-11-28
> **Looking Forward to Your Feedback**
>
> Dear reviewer wzb7,
>
> We sincerely appreciate the time and effort you’ve dedicated to reviewing our paper. As the discussion period ends soon on December 3rd, we wanted to kindly check if our responses have adequately addressed your concerns. If there are any remaining questions or points you'd like us to clarify, we would be more than happy to provide further details. On the other hand, if our rebuttal has resolved your concerns, we would be truly grateful if you could reflect that in your final evaluation.
>
> Thank you again for your thoughtful review, and we look forward to your feedback.
>
> Sincerely, The Authors

---

### Official Review · Reviewer_NrDF · 2025-11-10

**Soundness:** 3
**Presentation:** 4
**Contribution:** 4
**Rating:** 6
**Confidence:** 4

**Summary:**

This paper presents a mechanistic interpretability study of Video Large Language Models (VideoLLMs) to understand how they perform temporal reasoning for video question answering (VideoQA). Using techniques like Attention Knockout and Logit Lens, the authors trace the flow of information through the model's layers. They identify a consistent, multi-stage process where the model does temporal encoding, followed by video-language integration and finally answer formulation.
To validate their findings, the authors demonstrate that by preserving only these identified "effective information pathways" and pruning a substantial portion of the attention edges (e.g., 58% in LLaVA-NeXT-7B), the model's performance on VideoQA benchmarks remains largely intact, whereas random pruning of the same number of edges causes a significant performance drop. The study's findings are shown to be consistent across different model architectures and scales.

**Strengths:**

**Important and Timely Problem**: While VideoLLMs are advancing rapidly, their internal reasoning mechanisms are poorly understood. There is a large body of work done on LLMs, and somewhat less so for Vision Language Models (VLMs) (although recently has gathered much attention). This paper addresses a critical gap by providing one of the first mechanistic analyses of temporal reasoning in these models, moving beyond the more common studies on static image-based VLMs.

**Rigorous and Causal Methodology**: The authors employ established and powerful interpretability techniques (Attention Knockout, Logit Lens). The use of Attention Knockout is particularly strong, as it allows for causal interventions, disabling specific information pathways and measuring the direct impact on output probabilities. This provides much stronger evidence than purely observational methods like attention visualization.

**Clear Narrative and Strong Empirical Support**: The paper presents a compelling and easy-to-follow "blueprint" of information flow. Each stage of the proposed process is backed by targeted experiments. For instance, the comparison between image- and video-tuned models clearly demonstrates the emergence of cross-frame reasoning (Figure 2), and the analysis of information flow to question vs. option tokens provides a nuanced view of integration points (Figures 6 & 7).

**Thoroughness and Generalization**: The authors validate their findings across multiple modern VideoLLMs (LLaVA-NeXT 7B & 13B, Mini-InternVL-4B, VideoLLaMA3-7B) and datasets (TVBench, TOMATO). This demonstrates that the identified information flow patterns are a general property of current VideoLLMs rather than an idiosyncrasy of a single model.

**Weaknesses:**

**Reliance on Attention Knockout**: The primary evidence for this model of information flow is based on attention knockout, which is conceptually sound. However, this makes the takeaways prone to method-specific idiosyncrasies, so having an experiment to validate the conclusions which isn’t knocking out pathways for effective flow would strengthen the paper significantly.

For example, complex reasoning might involve more iterative processes, where later layers revisit or re-contextualize information from earlier layers (e.g., text representations directing further attention to specific video frames). The current knockout methodology, which blocks entire pathways across layers, may not capture such dynamic or recursive patterns. Blocking i->j for some layers but not all risks leaking alternative pathways of obtaining such information, which captures a flawed picture of what is happening. One example of this is Table 2, where a total cross-frame attention knockout from L1-16 could be extended to looking at the L1-N, N=1,...,32, to see when the integration is important until.

**Limited Scope of Tasks Format for Reasoning**: The analysis is primarily focused on multiple-choice VideoQA on the TVBench benchmark, which features relatively short videos and structured questions. It is unclear if this clean, sequential "blueprint" holds for more complex, long-form video tasks such as summarization, script generation, or open-ended dialogue, where temporal relationships are more complex and reasoning is less anchored to specific keywords. How are current pathways measured for multiple output token generations? Could such a method hold up for longer generations?

**Coarse-Grained Analysis of Representations**: The paper successfully identifies where and when information flows (i.e., which layers and between which token groups) but offers less insight into the substance of the information itself. For example, while Logit Lens shows the emergence of "temporal concepts" as vocabulary items, it doesn't reveal what features in the hidden states (e.g., motion, object state change, event boundaries) constitute these concepts. The analysis remains at the level of token interactions rather than diving into the underlying neural representations. The logit lens analysis could also be used to understand the spatial distribution of how the model is distributing attention, much like previous VLM works [1,2]. Something to this effect as an addition to this paper would be significant.

[1] Jiang et. al. Interpreting and Editing Vision-language Representations to Mitigate Hallucinations, ICLR 2025.

[2] Chen et. al., Why Is Spatial Reasoning Hard for VLMs? An Attention Mechanism Perspective on Focus Areas, ICML 2025.

**Questions:**

* The analysis convincingly shows that the "true option" token is a key integration point in multiple-choice QA. In a purely open-ended generation task without explicit options, where does this critical video-language integration converge? The paper suggests it shifts to the last token, but is the process as effective and localized without the explicit semantic anchors provided by the options?
* The pruning experiment is performed post-hoc on a trained model. Do you think these findings could be leveraged during training? For instance, could a regularization technique that encourages this sparse, efficient information flow lead to more robust, generalizable, or computationally efficient VideoLLMs? For example, if visual integration is important to occur early (as also observed in [3]) and is useful, can the embeddings be fed back in earlier?
* How do you hypothesize these pathways would adapt to videos where the crucial temporal information is not uniformly distributed, such as in "needle-in-a-haystack" tasks common in long-video understanding? Would the cross-frame interactions in early layers become more dynamic or would the model rely on different mechanisms entirely?
* The study focuses on temporal reasoning. Did you perform any analysis on tasks that are purely spatial (e.g., "what color is the object on the left throughout the video?") to see if the information pathways differ significantly, perhaps by bypassing the cross-frame interaction stage (see above comment on spatial distribution)?
* In Figure 1(b), why does Question -/-> Last have an increase in probability for the later layers?

[3] Nikankin et. al., Same Task, Different Circuits: Disentangling Modality-Specific Mechanisms in VLMs. arXiv 2025.

---

> ### Author Response · Authors · 2025-11-22
> **Author Response to Reviewer NrDF (1/4)**
>
> We sincerely thank Reviewer NrDF for the thoughtful comments and constructive suggestions. We appreciate the positive feedback highlighting our paper’s strengths, including the importance of the problem, the rigor of our methodology, the clarity of our narrative and empirical support, and the thorough experimental validation across multiple models and datasets.
>
> We first list up the modified part in the revised paper corresponding to each comment:
> * **W1. Reliance on Attention Knockout**
>     * Section F.4 "Signal leakage check" and Fig. V
> * **W2a & Q1. Further investigation on the information flow for open-ended questions**
>     * Section C.2 "Multiple Token Generation" and Figure S
> * **W2b. Further investigation on the information flow for complex reasoning & long-form videos**
>     * "Long video understanding" paragraph in Section F.2 and Fig. AB, AC, AD, and AE
> * **W3. More fine-grained analysis on concept emergence**
>     * Section D "Visualization of logit lens" and Figure T
> * **Q2, Future applications of our findings**
>     * "Future applications of our findings" paragraph in Section G
> * **Q3. Long-form videos with non-uniform temporal information**
>     * "Long video understanding" paragraph in Section F.2 and Fig. X, Y, Z, AA, AB, AC, AD, and AE
> * **Q4. Investigation on the spatial understanding task**
>     * "Spatial understanding" paragraph in Section F.2 and Fig. AF, AG, AH, and AI
> * **Q5. Clarification on probability increases in the last layers**
>     * "Discussion on the probability increases in the last layers" paragraph in Section G
>
> Below, we provide detailed responses to each point.
>
>
>
> > ### **W1. Concern about the reliance on attention knockout**
>
> We agree with your concern that our conclusions could be affected by relying on attention knockout. In response, we conducted the recommended experiment in the comment, and we are additionally designing analyses to validate our claim without relying on attention knockout during the remaining discussion period.
>
> **Gradual attention knockout on cross-frame interaction.**
> Following your advice, we extend the range of attention knockout.
> Specifically, instead of blocking cross-frame interaction only in the first half of the layers, we gradually block them from the first layer to the N-th layer for N=1,...,32. Moreover, we also conduct the reverse intervention by blocking from the last layer. This gradual intervention enables checking whether auxiliary signals from video tokens propagate beyond the mid layers, which would fall outside the identified range of effective information flow pathway for video tokens. As shown in Fig. V, the progressive knockout does not cause additional significant performance drop unless the intervention overlaps with the effective cross-frame interaction range we identified. This result further supports that our findings on cross-frame interaction are valid.
>
> **Additional analysis beyond attention knockout.**
> To further alleviate the reliance on attention knockout, we are also exploring alternative ways to identify effective information flow pathways without using attention knockout. We will update these results within the discussion period.

---

> ### Author Response · Authors · 2025-11-22
> **Author Response to Reviewer NrDF (2/4)**
>
> > ### **W2a & Q1. Further investigation on the information flow for open-ended questions**
>
> Thank you for the valuable comment. We agree that verifying the potential generalizability of our observations to open-ended VideoQA is important. Therefore, beyond analyzing the isolated case of generating a single temporal vocabulary token in Section C.1, we further investigate how the information flow is affected as the response is generated progressively when solving open-ended questions.s. Building on our findings in Section C.1 that the last token serves as the key checkpoint for video-language integration, we analyze how information flows among the video, question, generated response, and the last position evolves as generation proceeds.
>
>
> **Experimental setup.**
> We adopt the Temporal QA subset of VCGBench, a video conversation benchmark that includes diverse reasoning-based QA examples with different question types. While automatically identifying temporal vocabulary in generated responses is challenging, verbs serve as strong candidates, since they often contain action and time-related semantics crucial for solving VideoQA tasks. To extract temporal vocabulary from model responses, we employed spaCy's en-core-web-lg model to detect verbs and used their token positions as semantic anchors for our analysis.
>
> Then, we trace the probability change after Attention Knockout at each stage of anchor generation. For instance, given the question "What is happening in this video?'' with baseline response "A boy swings a bat and runs to the bases...", the detected anchors are \[swings, runs, ...\]. We then analyze information flow at different generation stages: generating the first anchor with no prior context (e.g., prompt: "USER: \<video\> What is happening in this video? ASSISTANT: A boy", target: "swings"), generating the second anchor after seeing one anchor (e.g., prompt: "USER: \<video\> What is happening in this video? ASSISTANT: A boy swings a bat and", target: "runs"), and so on.
>
> **Results.**
> Figure S depicts the effective information flow of different routes: video → question → last (a-b),  video → last (c), and video → response → last (d-e), as we vary the number of anchors from 1 (i.e., the experimental setup in Section C.2) to 2 and 3. Our results show that, as generation continues, newly produced temporal verbs in the response increasingly function as additional core checkpoints. This is evidenced by a clear shift in the dominant sources feeding the last position. As the number of temporal anchors increases, the contribution of response to last consistently grows, while the relative importance of video to last and question to last decreases (Fig. S(a), (b), (c)). We also observe a structural change in the route through which video evidence reaches the final prediction. When the number of anchors is 1, the model relies more on video to question to last, whereas with a larger number of anchors, it increasingly depends on video to response to last (Fig. S(d), (e)). We observe consistent monotonic trends as the number of anchors increases from 1 to 3, and therefore expect similar patterns to hold for more anchors. Overall, open-ended generation exhibits the same checkpoint-driven reasoning pattern observed in multi-choice QA. The model dynamically forms new checkpoints around temporal verbs, and effective information flow reorganizes accordingly, confirming that our core claims generalize to open-ended VideoQA without fixed answer options.
>
>
>
> > ### **W2b. Further investigation on the information flow for complex reasoning & long-form videos**
>
> We have also extended our analysis to long-form videos. We have validated VideoLLaMA3 on LongVideoBench, a video reasoning benchmark featuring complex, longer videos with tasks requiring sophisticated reasoning such as event localization. We have adopted Object-referred Event (O2E), Object before/after Object (O3O), and Scene-referred Object Tracking (SOS). We used 24 frames with 12x12 tokens per frame. As shown in Fig. AB, AC, AD, and AE, the information flow pathways for long-form VideoQA maintain similar patterns to short video benchmarks (Fig. K, L, M, and N). A notable difference is that the probability drop from cross-frame attention and video-to-question components is relatively smaller compared to short video benchmarks. We conjecture this to two factors: (1) long video benchmarks typically don't require every frame to be equally informative, reducing the need for comprehensive visual processing, and (2) questions in long video benchmarks contain more descriptive information, causing the model to rely more heavily on textual cues from the question rather than visual content.

---

> ### Author Response · Authors · 2025-11-22
> **Author Response to Reviewer NrDF (3/4)**
>
> > ### **W3. More fine-grained analysis on concept emergence**
>
> Thank you for the valuable comment. To further investigate the emergence of temporal concepts in videos, we visualize the token positions corresponding to spatial / temporal concepts using Logit Lens [1]. As shown in Fig. T, static concepts emerge first, and temporal concepts appear later as layers go deeper. Moreover, the emergent positions of temporal concepts spatially align with their relevant foreground regions; for example, the concept “sit” aligns with the region around a seated person. This result is consistent with our finding on concept emergence in video tokens in Section 3.3. Beyond this overall trend, we additionally observe that spatial concepts tend to settle on salient regions early, and temporal concepts then emerge mainly on the remaining tokens rather than replacing already stabilized spatial tokens. We interpret this behavior as a consequence of priority in spatial localization. Specifically, foreground regions are first mapped with spatial concepts that describe salient entities or attributes, and temporal concepts tend to emerge afterward. We hypothesize that this order of emergence enforces temporal concepts to occupy the remaining token positions not already taken by spatial concepts, which explains the positioning mechanisms of temporal concepts. To conclude, these visualizations extend our findings on concept emergence to a regional perspective.
>
> [1] nostalgebraist. Interpreting GPT: The logit lens. https://www.lesswrong.com/posts/ AcKRB8wDpdaN6v6ru/interpreting-gpt-the-logit-lens, August 2020.
>
>
>
> > ### **Q2. Future applications of our findings**
>
> Yes, we expect our findings can be leveraged during training of VideoLLMs. Current VideoLLMs often use only a limited part of the model to propagate information. Therefore, we may expand the utility of the model for such propagation by intentionally blocking the currently dominant information pathways during training. This encourages the VideoLLM to utilize alternative routes for information propagation, which can further exploit the potential capacity of the model. Our findings can also be leveraged for testing. Our findings suggest that we can apply early-exiting [2,3,4] to tokens that lie beyond the range of effective information flow pathways, which may reduce the overall computation cost without significantly affecting accuracy.
>
> [2] Maha Elbayad, Jiatao Gu, Edouard Grave, and Michael Auli. 2020. Depth-adaptive transformer. In 8th International Conference on Learning Representations, ICLR 2020
>
> [3] Tal Schuster, Adam Fisch, Jai Gupta, Mostafa Dehghani, Dara Bahri, Vinh Tran, Yi Tay, and Donald Metzler. Confident adaptive language modeling. Advances in Neural Information Processing Systems, 2022
>
> [4] Sangmin Bae, Jongwoo Ko, Hwanjun Song, Se-Young Yun, Fast and Robust Early-Exiting Framework for Autoregressive Language Models with Synchronized Parallel Decoding, EMNLP 2023
>
>
>
> > ### **Q3. Long-form videos with non-uniform temporal information**
>
> Thank you for the insightful question. To address the case where crucial temporal information is not uniformly distributed, we extend our analyses to a long-form VideoQA setting, where crucial information is more sparse.
> To this end, we evaluate VideoLLaMA3 on LongVideoBench, which contains longer and more complex videos. We used Object referred Event (O2E), Object before after Object (O3O), and Scene referred Object Tracking (SOS), which require identifying information from specific moments within long-form videos. For example, the O3O task includes questions such as "What is the first concept mentioned after the man, sitting in front of the microphone wearing a black shirt with a pattern on the neck and a black cap and black-rimmed glasses, talks about evolution?", which matches the concerned point by requiring the retrieval of a specific temporal segment. Here, we adopt both 8 and 24 frame sampling to check a more sparse distribution of evidence under 24 frames. As shown in Fig. X, Y, Z, AA, the same effective information flow pathways largely persist in this long video setting, indicating that analysis results for cross-frame interactions and video-language integration appear consistent with our previous analysis on TVBench (Fig. L, M, N, and O). Moreover, Fig. AB, AC, AD, and AE show that the results under 24 frames follow a similar trend to 8 frames, indicating that the model does not rely on the length of the input videos. These results validate that our findings remain applicable to long-form videos with non-uniformly distributed crucial temporal information.

---

> ### Author Response · Authors · 2025-11-22
> **Author Response to Reviewer NrDF (4/4)**
>
> > ### **Q4. Investigation on spatial understanding VideoQA task**
>
> Following the comment, we extend our analysis to VideoQA for spatial understanding tasks. To this end, we investigate two tasks of the Video-MME benchmark: the action recognition task and the spatial perception task.
> As shown in Figure AF, when we block the cross-frame interaction, the action recognition task exhibits a clear and consistent performance drop, indicating that temporal aggregation is crucial for this setting. In contrast, the spatial perception task shows a much smaller average degradation and a much larger variance across samples. We conjecture that this pattern arises because some spatial perception questions incidentally benefit from temporal information, whereas many others can be answered from static scenes. Therefore, the impact of blocking cross-frame interaction ranges from almost no change to a significant drop. Overall, this analysis supports our claim that spatial understanding VideoQA tasks are less sensitive to cross-frame interaction.
>
>
>
>
> > ### **Q5. Clarification on the probability increases in the last layers**
>
> By interleaving “the investigation of information flow from the question to the last token” (Fig. 3 and B) with “the answer generation behavior in the middle-to-late layers” (Fig. 8 and E), we can observe that when information is actively propagated from the question to the last token, the probability of the true option tends to increase. This is because major information flow from the question to the last token mainly helps the VideoLLM consolidate evidence for the true option, so the increase in probability is beneficial. However, by the time we reach the late layers, the model already has a relatively stable belief that it should choose the true option.  In this regime, keeping the pathway from the question to the last token fully open until the final layer can act as an additional amplifier for all option-related signals, including those associated with false options, which raises the probabilities of both true and false options. Therefore, when we block this pathway only in the final layers, the model still relies on the information that has already been propagated and stored in the hidden states, so the evidence for the true option remains. At the same time, the absence of further propagation prevents additional amplification of the false options. As a result, the probability of the true option can increase relative to the false options, and in some cases even increase in absolute terms, which explains the behavior observed in Fig. 1(b).

---

> ### Author Response · Authors · 2025-11-28
> **Looking Forward to Your Feedback**
>
> Dear reviewer NrDF,
>
> We sincerely appreciate the time and effort you’ve dedicated to reviewing our paper. As the discussion period ends soon on December 3rd, we wanted to kindly check if our responses have adequately addressed your concerns. If there are any remaining questions or points you'd like us to clarify, we would be more than happy to provide further details. On the other hand, if our rebuttal has resolved your concerns, we would be truly grateful if you could reflect that in your final evaluation.
>
> Thank you again for your thoughtful review, and we look forward to your feedback.
>
> Sincerely, The Authors

---

### Author Response · Authors · 2025-12-04
**Author Final Remark (1/2)**

We sincerely thank the SAC, AC, and reviewers for their efforts and valuable feedback throughout the review process.

In this paper, we provide a comprehensive blueprint of VideoLLM internal mechanisms, revealing where and how these models extract and propagate video and textual information in VideoQA tasks. Through mechanistic interpretability techniques including Attention Knockout, Logit Lens, and attention visualization, we demonstrate that VideoLLMs follow a structured three-stage process: active cross-frame interactions in early-to-middle layers, video-language integration with aligned keywords in middle layers, and answer completion in late layers. These findings are validated across different model architectures, scales, and datasets.

We are pleased that our work has been commended for the following aspects:
* **Research Significance**: Understanding internal information flow in VideoLLMs addresses an important yet under-explored problem (NrDF, trVM, 5oZh) and represents an interesting and novel research topic (wzb7)
* **Methodological Rigor**: Rigorous experimental design and systematic application of mechanistic interpretability techniques (NrDF, 5oZh)
* **Clarity and Structure**: Clear narrative with an easy-to-follow presentation (NrDF, 5oZh)
* **Practical Impact**: Provides a comprehensive "blueprint" of information flow (NrDF) that delivers valuable insights to the research community (trVM, 5oZh)
* **Robust Validation**: Thorough experimental evaluation across multiple models and datasets demonstrates scalability and generalization (NrDF, wzb7)

---

> ### Author Response · Authors · 2025-12-04
> **Author Final Remark (2/2)**
>
> During the discussion phase, we thoroughly addressed all reviewer concerns through additional investigations as follows:
>
> * **Open-ended QA generalization (NrDF, trVM)**
>   * Concern: How do observations on video-language integration with aligned keywords generalize to open-ended QA with multiple token generations, where the model generates answers without given fixed options in question?
>   * Response: We extended our analysis to open-ended QA benchmark (VCGbench). Building on our previous finding that the last token serves as the key checkpoint for video-language integration in open-ended generation, we further examined how checkpoints emerge within responses and how information propagates through these checkpoints as the model continues to generate multiple tokens. We observed the model dynamically forms new checkpoints around temporal verbs and effective information flow reorganizes accordingly. This confirms that our previous finding in single token generation at open-ended QA generalizes to multiple token generations with longer responses.
>
> * **Concept emergence mechanisms (NrDF, wzb7)**
>   * Concern: Need deeper analysis on concept emergence (e.g., why spatial and temporal concept timelines differ)
>   * Response: We extended our analysis to a regional perspective by visualizing token positions of emerged spatial/temporal concepts using Logit Lens. Beyond the original observation of different concept emergence timelines, we additionally found spatial concepts settle on salient regions early, then temporal concepts emerge mainly on remaining tokens rather than replacing already stabilized spatial tokens. We hypothesize that this order of emergence enforces temporal concepts to occupy the remaining token positions not already taken by spatial concepts, which explains the positioning mechanisms of temporal concepts. Overall, this analysis provides a more concrete explanation of the emergence order of spatial and temporal concepts and their effect on where temporal concepts are grounded.
>
> * **Additional attention map visualizations (5oZh)**
>   * We added video-to-question attention map examples for all tasks, confirming information flow from video to temporal keywords is significantly impacted by their alignment built through cross-frame interactions.
>
> * **Long-form video generalization (NrDF, trVM, 5oZh)**
>   * We validated VideoLLaMA3 on LongVideoBench with varying frame lengths (8 frames, 32 frames), confirming our observations hold for longer videos.
>
> * **Spatial vs. temporal understanding tasks (NrDF, 5oZh)**
>   * To compare how do spatial vs. temporal understanding tasks affect information flow, we added action recognition and spatial perception tasks in Video-MME benchmark. Overall effective flow pathways remain consistent; only the dominance of cross-frame interactions on final performance differs between task types.
>
> * **Failure case analysis (trVM)**
>   * Concern: In failure cases, are model errors due to breakdown in the same pathways with success case?
>   * Response: We analyzed probability changes for wrong answers in failure samples. We found cross-modal flow patterns are consistent with the successful case, indicating root causes lie in earlier video representation stages. Cross-frame intervention revealed two failure patterns: 1) model being overconfident in false options from cross-frame interaction or video-language misalignment; 2) model over-relies on per-frame static scenes.
>
> * **Attention knockout validity (NrDF)**
>   * Concern: Reliance on attention knockout methodology of blocking entire pathways across layers may not capture potential signal leakage.
>   * Response: We added Attention Knockout experiment by progressively blocking from the first layer to the last layer, and the last layer to the first layer. The results are consistent with the previously identified effective cross-frame interaction ranges, further validating our findings.
>
> * **Future applications of our findings (NrDF, 5oZh)**
>   * We discussed applicable ways for training and testing: 1) activating currently rarely-used model pathways during training and 2) improving efficiency through early-exiting strategies during testing.
>
> * **Probability increases clarification (NrDF)**
>   * Reviewer raised question on why does Question -/-> Last show probability increases in later layers. We discussed that blocking the pathway only at the final layers preserves earlier propagated evidence for the true option in the hidden states while preventing further amplification of false options, making the true option probability to rise relatively.
>
> * **Missing relevant studies (5oZh)**
>   * We included the missing relevant studies (e.g., video reasoning benchmarks) in the related work section.
>
> We thank all reviewers for their valuable and insightful questions, which helped us significantly improve the paper. We believe our work offers practical insights that will benefit the broader research community in understanding and improving VideoLLMs.

---

### Meta-Review · Area_Chair_e4cv · 2026-01-03

**Summary:**

The reviewers generally agreed that this paper addresses a critical and timely problem: understanding the internal "black box" mechanisms of VideoLLMs during temporal reasoning. The initial positive consensus was driven by the methodological rigor of using causal interventions (Attention Knockout) and the discovery of a consistent three-stage "blueprint" for information flow across different model architectures.

The reviewers also raised some concerns: 1) whether the observed pathways were merely artifacts of multiple-choice shortcuts, 2) if the findings would hold for long-form videos beyond the 8-frame setup, and 3) the lack of analysis regarding failure modes (why models give wrong answers). The recommendation for acceptance is based on the authors' exceptionally thorough empirical response to these questions.

**Reviewer Concerns:**

Concerns Addressed by the Rebuttal
* Generalization to open-ended tasks (`NrDF`, `trVM`): The reviewers were concerned that the "integration checkpoints" were shortcuts for verifying fixed options induced by the multiple-choice QA formulation. In response, the authors extended their analysis to open-ended generation (VCGBench), showing that the model dynamically forms new checkpoints around temporal verbs. This suggests the mechanism not just multiple-choice verification but rather fundamental to language generation.

* Analysis limited to 8-frame videos (`trVM`, `5oZh`): Concerns that an 8-frame window was too short were resolved through new experiments on LongVideoBench (using 24–32 frames), showing the three-stage pattern persists even in longer sequences.

* Failure mode analysis (`trVM`): The authors added a mechanistic study of incorrect predictions. They found that errors often stem from over-reliance on static frames in the early representation stage rather than a total collapse of the integration pathways, providing a more complete picture of model behavior.

* Signal leakage and knockout validity (`NrDF`): The concern that blocking entire pathways across all layers might miss recursive patterns was addressed via progressive knockout experiments (blocking L1 to LN).

Outstanding Concerns
* Fine-grained feature interpretation (`NrDF`, `wzb7`): While the authors provided regional Logit Lens visualizations to show where concepts emerge, the analysis still remains at the token level. A deeper understanding of what specific features (e.g., optical flow vs. object edge changes) constitute these hidden states remains an open question.

**Reviewer Scores:**

* Reviewer `wzb7` (Initial: 8 $\rightarrow$ Estimated: 8): Already very positive; the regional Logit Lens analysis directly addressed their request for a deeper "why" regarding concept emergence.

* Reviewer `5oZh` (Initial: 8 $\rightarrow$ Estimated: 8): Maintained their high score after explicitly stating in the discussion that the comprehensive response "effectively addressed earlier concerns and further strengthened the paper."

* Reviewer `NrDF` (Initial: 6 $\rightarrow$ Estimated: 7): This reviewer provided the most rigorous technical critique. The success of the progressive knockout and open-ended verb tracing addressed their primary theoretical and methodological doubts.

* Reviewer `trVM` (Initial: 6 $\rightarrow$ Estimated: 7): The failure case analysis and long-video generalization were the specific "missing pieces" requested by this reviewer; their resolution likely moves this to a full Accept.

---

### Decision · Program_Chairs · 2026-01-26

Accept (Poster)